# Myomerger promotes fusion pore by elastic coupling between proximal membrane leaflets and hemifusion diaphragm

Gonen Golani[1,5], Evgenia Leikina[2,5], Kamran Melikov[2], Jarred M. Whitlock [2], Dilani G. Gamage[3], Gracia Luoma-Overstreet[2], Douglas P. Millay [3,4], Michael M. Kozlov [1✉] & Leonid V. Chernomordik [2✉]

Myomerger is a muscle-specific membrane protein involved in formation of multinucleated muscle cells by mediating the transition from the early hemifusion stage to complete fusion. Here, we considered the physical mechanism of the Myomerger action based on the hypothesis that Myomerger shifts the spontaneous curvature of the outer membrane leaflets to more positive values. We predicted, theoretically, that Myomerger generates the outer leaflet elastic stresses, which propagate into the hemifusion diaphragm and accelerate the fusion pore formation. We showed that Myomerger ectodomain indeed generates positive spontaneous curvature of lipid monolayers. We substantiated the mechanism by experiments on myoblast fusion and influenza hemagglutinin-mediated cell fusion. In both processes, the effects of Myomerger ectodomain were strikingly similar to those of lysophosphatidylcholine known to generate a positive spontaneous curvature of lipid monolayers. The control of post-hemifusion stages by shifting the spontaneous curvature of proximal membrane monolayers may be utilized in diverse fusion processes.

---

[1] Department of Physiology and Pharmacology, Sackler Faculty of Medicine, Tel Aviv University, Tel Aviv 69978, Israel. [2] Section on Membrane Biology, Eunice Kennedy Shriver National Institute of Child Health and Human Development, National Institutes of Health, Bethesda, MD 20892, USA. [3] Division of Molecular Cardiovascular Biology, Cincinnati Children's Hospital Medical Center, Cincinnati, OH 45229, USA. [4] Department of Pediatrics, University of Cincinnati, Cincinnati, OH 45229, USA. [5] These authors contributed equally: Gonen Golani, Evgenia Leikina. ✉email: michk@tauex.tau.ac.il; chernoml@mail.nih.gov

Fusion of two membrane bilayers is a key event in the entry of enveloped viruses, in fertilization, development and remodeling of muscles and bones, and in intracellular membrane trafficking (reviewed in Brukman et al.[1]). Diverse membrane fusion processes apparently converge on a common pathway of lipid rearrangements that can be divided into three steps. First, merger of the proximal monolayers of the two opposing bilayers into a hemifusion connection, referred to as the fusion stalk, establishes continuity between these monolayers and enables lipid mixing between the membranes. Second, an expansion of the stalk brings the distal lipid monolayers of the bilayers together into a hemifusion diaphragm (HD). Third, opening and expansion of a fusion pore within the HD allows for content mixing between the volumes surrounded by the membranes. Different stages of the fusion pathway involve membrane deformations and depend on the membrane leaflets propensity to bend[2,3]. Theoretical analysis and experiments on fusion of protein-free lipid bilayers and different biological fusion processes indicate that addition of lipids with an effective molecular shape of inverted-cones such as lysophosphatidylcholine (LPC), also referred to as the lipids of positive spontaneous curvature, to the proximal monolayers of the fusing bilayers inhibits hemifusion[2]. Inhibition of hemifusion by LPC has been used in our earlier studies to uncouple actual fusion events in myoblast differentiation from its pre-fusion stages[4,5].

Myoblast fusion, a key process in the development and regeneration of skeletal muscle, is one of the best characterized examples of cell–cell fusion. A number of proteins and lipids have been implicated in myoblast fusion (reviewed in refs. [6–9]). Two muscle-specific proteins have been discovered to be essential for myoblast fusion: Myomaker[10] and Myomerger/Myomixer/Minion[11–13]. Myomaker is required for hemifusion[5] whereas the subsequent transition from hemifusion to complete fusion depends on Myomerger. The extracellularly localized region (ectodomain) of Myomerger drives opening of a fusion pore, as demonstrated for myoblast fusion and for heterologous fusion reaction mediated by the influenza virus hemagglutinin (HA)[5].

The role of Myomerger in myoblast fusion represents a paradigm applicable to all fusion events, the essence of which is the promotion of fusion pore formation in the HD that consists of the distal (inner) monolayers of the fusing membranes by proteins/protein regions that do not directly interact with these membrane monolayers. Indeed, many functionally important regions of the proteins that mediate fusion in viral infections and in developmental processes are located in their ectodomains and interact with proximal monolayers. Because of the location of the fusion proteins, most of the suggested mechanisms of protein-mediated fusion have explained only how these proteins drive merger of proximal monolayers resulting in hemifusion by bringing proximal monolayers into very tight contact and bending them[2,14,15]. Considering that Myomerger is not involved in hemifusion but drives the transition from hemifusion to fusion, this protein presents an important model for analysis of the physical mechanisms underlying an, apparently, indirect effect of Myomerger and other proximal leaflet-associated factors on the fusion pore formation.

In this study we explored how Myomerger promotes hemi-to-full fusion transition in myoblast fusion. We hypothesized that Myomerger shifts the spontaneous curvature of the proximal membrane monolayers towards positive values. This shift generates additional tension in the distal monolayers composing the HD and, hence, promotes the fusion pore formation. In support of our hypothesis, we found experimentally that a synthetic polypeptide sMyomerger[26-84] with an amino acid sequence corresponding to Myomerger protein lacking the N-terminal transmembrane domain of the protein (amino acids 1–25)[5] generates

positive spontaneous curvature of lipid monolayer. We then theoretically analyzed the effects of the positive spontaneous curvature of proximal membrane monolayers on the elastic stresses in HD. Our modeling revealed an HD rim mediated elastic crosstalk between the proximal membrane monolayers and the HD. The crosstalk elicits growing HD tension and fusion pore formation following increased proximal monolayer spontaneous curvature. Our experiments verified this key prediction of the theoretical model by demonstrating that the fusion defect in Myomerger-deficient myoblasts can be partially rescued not only by application of sMyomerger[26-84] but also by application of LPC, a lipid of positive spontaneous curvature. Moreover, the effects of sMyomerger[26-84] and LPC on HA-mediated cell fusion were strikingly similar.

As expected based on the earlier work[2], a sufficiently strong shift of the spontaneous curvature in the proximal monolayers to positive values by LPC or, as we found here, by sMyomerger[26-84] inhibited hemifusion and, consequently, fusion. However, for both myoblast fusion and HA-mediated fusion, the concentrations of sMyomerger[26-84] and LPC that promoted hemifusion-to-fusion transition were considerably lower than those required for the hemifusion inhibition. We suggest that levels of Myomerger expression characteristic for fusion-committed myoblasts are in the range allowing Myomerger to promote rather than inhibit myoblast fusion. Our estimate for the surface density of sMyomerger[26-84] that rescues fusion of Myomerger-deficient cells is comparable to the surface densities reported for viral fusogens. The physical mechanism, by which, according to our analysis, Myomerger drives fusion pore opening, can also underlie the effects of other proximal leaflet-associated factors on the fusion pore formation in diverse fusion processes.

## Results

**Myomerger generates a positive contribution to the spontaneous curvature of lipid monolayers.** In Leikina et al.[5], we reported that Myomerger facilitates formation of pores in cell membranes and liposomes and that the fusion defect in Myomerger-deficient myoblasts can be partially rescued by application of a pore-forming peptide magainin 2. Because of a positive curvature of the lipid monolayer at the edge of a lipidic pore[2] and the known ability of magainin 2 to impose positive curvature strain[16], we suggested that Myomerger interactions with the lipids in the proximal leaflets of the plasma membranes induce positive spontaneous curvature of these lipid monolayers[5].

To test this hypothesis, we developed an experimental approach based on a very strong dependence of fluorescence resonance energy transfer (FRET) from a donor dye molecule to an acceptor dye molecule without emission of a photon on a donor-to-acceptor distance. Amphipathic molecules such as phospholipids with inverted cone-like effective molecular shapes or alpha-helical peptides, which get shallowly inserted into lipid monolayers, produce positive contributions to the spontaneous curvature of the monolayers[17]. The background for this effect is an expansion of the monolayer region formed by the polar heads with respect to that filled by the hydrocarbon tails of lipid molecules[17]. Therefore, the amphipathic membrane insertions, which generate positive contributions to the monolayer spontaneous curvature are expected to increase the average donor-to-acceptor distance for FRET dyes bound to the polar heads of lipids stronger than for dyes placed at the ends of the lipid acyl chains.

We prepared large unilamellar liposomes from 9-to-1 mix of dioleoyl phosphatidylcholine (DOPC) and dioleoyl phosphatidylserine (DOPS) supplemented with either FRET pair of lipid head-placed probes (18:1 TopFluor PE and 18:1 Liss Rhod PE,

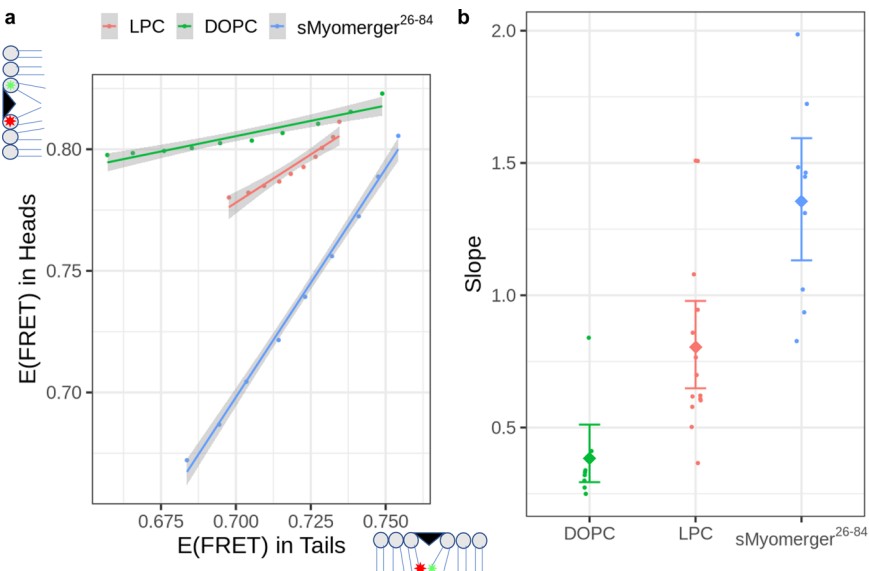

**Fig. 1 Effects of insertion of different molecules into lipid bilayer on efficiency of FRET between probes located in lipid headgroup and between probes in lipid hydrocarbon tail regions. a** Increasing bulk concentrations of either dioleoyl phosphatidylcholine, DOPC (from 1 μM to 10 μM), or lysophosphatidylcholine, LPC (from 10 μM to 100 μM) or sMyomerger[26-84] (from 50 nM to 0.5 μM) were added to 2.5 μM liposomes containing fluorescence resonance energy transfer (FRET) pair with both probes located in the lipid headgroup region or with both probes located in the lipid hydrocarbon tail region. Dependence of FRET efficiency characterized as change in quantum efficiency of donor emission due to the presence of acceptors (see Methods section) for probes in headgroup region E(FRET) in Heads vs E(FRET) in Tails for probes in hydrocarbon tails from a representative experiment is shown. For all three reagents and for probes both in the headgroup region and in hydrocarbon tails, E(FRET) monotonously decreased as the bulk concentration of the reagent increased. Individual points correspond to a single bulk concentration of the added molecules, the line shows linear fit of the data and error bands are centered on the linear fit values and show 0.95 confidence interval of the linear regression. **b** Slopes of the E(FRET) in Heads vs E(FFET) in Tails are different for DOPC, LPC, and sMyomerger[26-84] with DOPC < LPC < sMyomerger[26-84]. Individual dot points represent a slope estimate obtained as shown in **a** from an independent experiment (*n* = 14, 9, and 8 for LPC, sMyomerger[26-84], and DOPC, respectively), diamond points and error bars represent slope means with 95% confidence interval.

0.5 mol% each) or FRET pair of acyl chain-placed probes (TopFluor® PC and TopFluor® TMR PC, 0.5 mol% each). Application of lauroyl LPC to liposomes with lipid head-placed probes and to liposomes with acyl chain-placed probes decreased efficiency of FRET. We then carried out similar experiments with sMyomerger[26-84]. For different bulk concentrations of the insertions, we plotted the changes in the efficiency of FRET for probes placed at the lipid heads vs. the efficiency of FRET for probes in the tails (Fig. 1a). The higher the slope of these dependencies, the greater the ability of a given insertion to push apart the lipid heads as compared to its ability to separate the lipid acyl chains, which corresponds to splay of the lipid molecules. Such splay deformation is described by a positive spontaneous curvature of the lipid monolayer containing the insertions. The slopes of the head FRET/tail FRET dependencies for sMyomerger[26-84], DOPC and LPC from independent experiments are aggregated in Fig. 1b. Finding that Myomerger ectodomain pushes polar heads of the lipids apart stronger than LPC indicates that membrane-inserted Myomerger ectodomain generates an even larger positive contribution to the monolayer spontaneous curvature than LPC.

We hypothesized that the generation by Myomerger of a large positive contribution to the monolayer spontaneous curvature is a key factor for its role in cell–cell fusion. To test this idea and explore the underlying physical mechanism, we employed theoretical analysis.

**Physical mechanism.** Here we propose a mechanism of promotion of the hemifusion-to-fusion transition by the membrane insertions, which support the shift of the spontaneous curvature of the proximal leaflets of the fusing membranes to more positive values. The mechanism captures an indirect effect of the proximal membrane leaflets on the fusion pore generation in the HD, which is composed of the distal membrane leaflets. The essence of the mechanism is a geometrical and mechanical coupling of the proximal and distal leaflets of the fusing membranes mediated by the HD rim.

We consider the fusion site at the stage of hemifusion as consisting of HD surrounded by elements of the initial membranes. The fusion site has an axially symmetric configuration, whose half-cross-section is illustrated in Fig. 2a. The HD rim represents the region of junction between three membranes: the elements of the two initial membranes and the HD membrane. The structure of the HD rim can be characterized by the angle, $\varphi_0$, formed by the membrane mid-surfaces along the junction line as it appears in the cross-section of the structure (Fig. 2a). We will refer to $\varphi_0$ as the junction angle.

We consider the zone of the initial membrane opposition to be established by special proteins, which hold the two membranes parallel to each other. We define as the intermembrane distance, $x^*$, the spacing between the so-called neutral planes of the opposing proximal monolayers, which underlie the monolayer polar head regions[18,19] (Fig. 2a). In our analysis we varied $x^*$ in the range between 4.6 and 8.6 nm, which, after subtracting the thickness of two polar head layers of ~2 nm[20], is close to the 2.5–5 nm diameters of the ectodomains of the post-fusion hairpin conformations of the well characterized protein fusogens, which lay parallel to the membranes after bringing membrane bilayers together[21,22]. We assume that, in the course of hemifusion and fusion, the distance-establishing proteins form a ring-like zone around the fusion site. The internal radius of this zone, $R_B$, determines the radius of the fusion site boundary and dimension

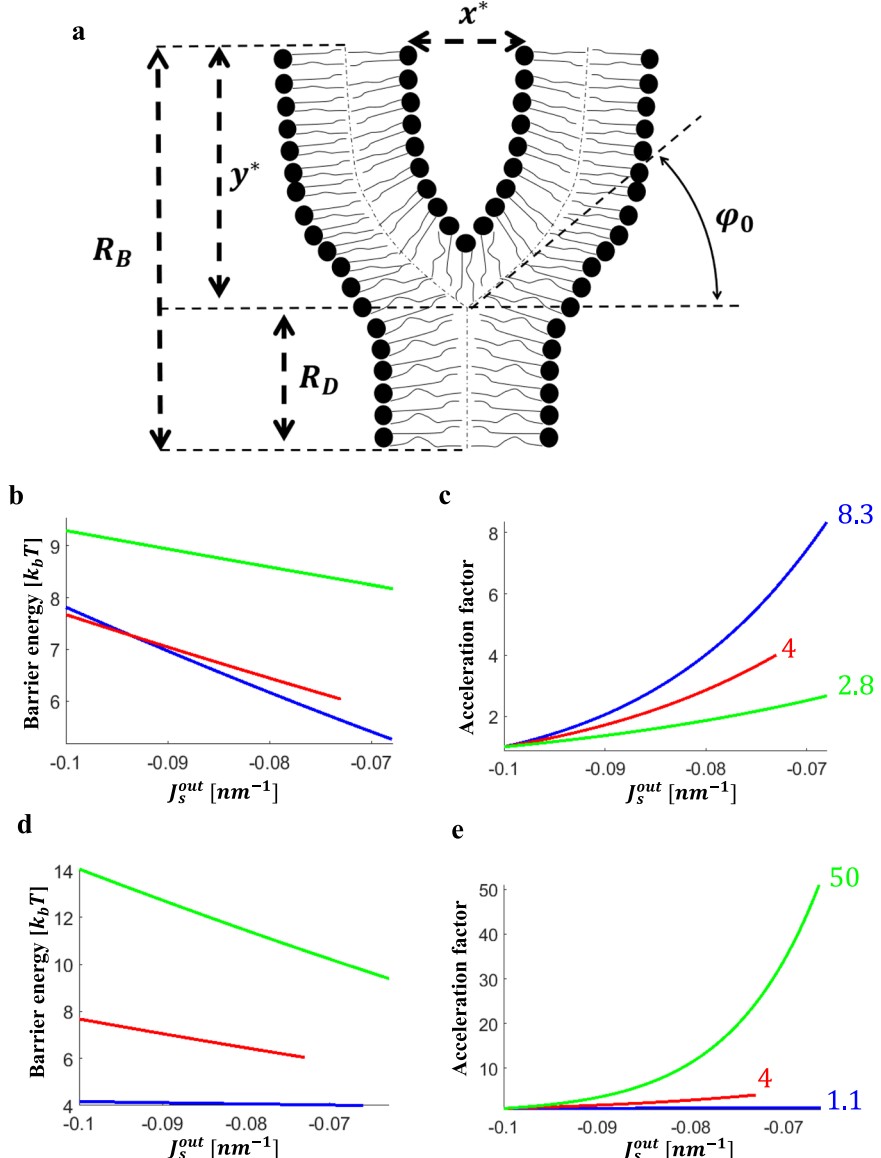

**Fig. 2 Effects of fusion site dimensions on the energy barrier and the acceleration factor of the fusion pore formation. a** Configuration of half-cross-section of the fusion site and notations. $\varphi_0$ - Junction angle, $R_D$- diaphragm radius, $R_B$- fusion site radius, $x^*$ - intermembrane distance, $y^*$ - the distance between the HD rim and the boundary of the fusion site. For simplicity, the relative dimensions are drawn out of scale. **b**, **c** Panels **b** and **c** describe the effects of varying the intermembrane distance, $x^*$, on the energy barrier (**b**) and the acceleration factor (**c**), as a function of spontaneous curvature of the proximal monolayers. Blue line $x^* = 4.6$ nm, red line $x^* = 6.6$ nm, and green line $x^* = 8.6$ nm. **d**, **e** Panels **d** and **e** describe the effects of varying fusion site radius, $R_B$, on the energy barrier (**d**) and the acceleration factor (**e**), as a function of spontaneous curvature of the proximal monolayers. Blue line $R_B = 10$ nm, red line $R_B = 15$ nm, and green line $R_B = 20$ nm. The parameter values used in all panels are $l = 1.5$ nm, $\kappa_m = 10 k_b T$, $\bar{\kappa}_m = -5 k_b T$, $\lambda = 10$ pN. The radius of the fusion site in **b** and **c** is $R_B = 15$ nm and the intermembrane distance in panels **d** and **e** is $x^* = 8.6$ nm. We indicate in **c** and **e** the acceleration factor for the maximal calculated LPC area fraction in the proximal monolayers, $\phi_{LPC}$. **c** Blue line, $x^* = 4.6$ nm, $\phi_{LPC} = 10\%$, $\beta = 8.3$. Red line, $x^* = 6.6$ nm, $\phi_{LPC} = 7.5\%$, $\beta = 4$. Green line, $x^* = 8.6$ nm, $\phi_{LPC} = 10\%$, $\beta = 2.8$. **e** Blue line, $R_B = 10$ nm, $\phi_{LPC} = 10\%$, $\beta = 1.1$. Red line, $R_B = 15$ nm, $\phi_{LPC} = 7.5\%$, $\beta = 4$. Green line, $R_B = 20$ nm, $\phi_{LPC} = 10\%$, $\beta = 50$.

of the HD is characterized by the radius of its circular rim, $R_D$ (Fig. 2a). While the variations of the rim radius, $R_D$, are not restricted, if not stated otherwise, we assume that the radius of the fusion site boundary, $R_B$, does not change.

Since HD is formed by the distal rather than proximal monolayers of the interacting membranes, the elastic properties of the latter can only indirectly influence the HD remodeling. We propose the existence of an elastic crosstalk between the proximal membrane monolayers and those of the HD, which is mediated by the HD rim. Qualitatively, upon variation of the proximal monolayer spontaneous curvature, $J_S^{out}$, the junction angle, $\varphi_0$,

and the diaphragm radius, $R_D$, must change to adjust the configurations of the junction region to the new value of $J_S^{out}$. Since the spontaneous curvature of the distal monolayers, $J_S^{in}$, remains unchanged, this can happen only at the expense of deviation of the distal monolayers from their optimal configurations within the junction region. This must generate additional elastic stresses in the distal monolayers, which propagate to some distance into the HD resulting in an extra lateral tension in the HD that accelerates the fusion pore formation. To test this hypothesis, we theoretically analyzed the effect of the spontaneous curvature of the proximal monolayers of the fusing

membranes, $J_S^{out}$, on the elastic stresses and the related rate of the fusion pore formation within the HD.

We determined the optimal configurations of the fusion site by computing the $\varphi_0$, and $R_D$ in a relevant range of $J_S^{out}$. For these configurations, we found the elastic energy accumulated within the fusion site (Appendix A) and the related distribution of the lateral tension, $\gamma$, within the HD as a function of $J_S^{out}$ (Appendix B). Based on the obtained results, we evaluated (Appendix C) the dependence of the rate of the pore formation in the HD on $J_S^{out}$ by computing the acceleration factor, $\beta$, which we define as a ratio between the characteristic times of the fusion pore formation in the initial state, $\tau_i$, and that after the $J_S^{out}$ variation, $\tau_f$, as

$$\beta = \tau_i / \tau_f \qquad (1)$$

The model equations as well as the methods used for their solution are presented in full detail in Appendices A, B, C.

The physical model underlying our computational analysis is based on the concept of membrane bending elasticity operating with the elastic moduli of lipid monolayers: the bending modulus, $\kappa_m$, the tilt modulus, $\kappa_t$, the saddle-splay, $\bar{\kappa}_m$ (see physical model in Appendix A), and the line tension of the fusion pore rim, $\lambda$. We assumed the bending modulus to adopt values within the experimentally measured range of $\kappa_m = 10–40$ k$_B$T (where k$_B$T $= 4.11 \times 10^{-21}$ Joule is the product of the Boltzmann constant and the absolute temperature)[23,24]. The values of the monolayer tilt, $\kappa_t$, and saddle-splay, $\bar{\kappa}_m$, moduli have not been directly measured but rather estimated based on indirect experimental data[25] and computations[26]. Therefore, we performed the calculations for these moduli varying within the ranges, which include the estimations currently available in the literature, $\kappa_t = 20 – 80$ mN m$^{-1}$ and $\bar{\kappa}_m = -40 – 0$k$_B$T[27–29]. These ranges of parameters correspond to the typical decay length of the tilt deformations, $l = \sqrt{\kappa_m/\kappa_t}$[26,30], varying in the range between 1 and 2 nm.

The bilayer thickness and the radius of the fusion site boundary, $R_B$ are taken to be 3 nm and 15 nm, respectively. The distance between the HD rim and the boundary of the fusion site, $y^*$ (Fig. 2a), is assumed to be much larger than the tilt decay length, $l$.

We assumed that the background spontaneous curvature, $J_S^0$, existing before the variations of the distal monolayer composition, is equal for the proximal and distal monolayers and has a value $J_S^0 = -0.1$ nm$^{-1}$, typical for lipids like phosphatidylcholines (PC), which are ubiquitous in cell membranes (measured in Szule et al.[31] for dioleoyl PC). We further assumed that the variations of the spontaneous curvature of the proximal monolayer, $J_S^{out}$, result from addition to this monolayer of molecules with positive effective molecular curvature, $\zeta$, which has a value of $\zeta = 0.26$ nm$^{-1}$ corresponding to the value measured for oleoyl lysophosphatidylcholine (LPC)[32]. The proximal monolayer spontaneous curvature, $J_S^{out}$, is assumed to be given by[33,34]

$$J_S^{out} = (1 - \phi)J_S^0 + \phi \cdot \zeta, \qquad (2)$$

where $\phi$ is the monolayer area fraction occupied by the added molecules (LPC). The spontaneous curvature of the distal monolayers remains equal to the background value, $J_S^{in} = J_S^0$.

Finally, the line tension of the fusion pore rim, $\lambda$, was assumed to have values in the range $\lambda = 10–30$ pN[35].

**Results of the modeling.** For each value of the proximal monolayer spontaneous curvature, $J_S^{out}$, corresponding, according to (Eq. 2), to a certain LPC area fraction, $\phi$, we computed the system configuration of the minimal overall elastic energy (Appendix A, Eq. A5) and found the corresponding distribution

of the lateral tension in the HD (Appendix B, Eq. A15). Based on that, we computed the energy barrier of the pore formation as a function of the spontaneous curvature, $E^*(J_S^{out})$ (Appendix C, Eq. A19). We computed the acceleration factor of the pore formation, $\beta$, from the difference between the energy barrier, $E_0^*$, in the initial situation of symmetric membranes, $J_S^{out} = J_S^{in} = J_S^0$, and that after the variation of the proximal monolayer spontaneous curvature, $J_S^{out}$, according to (Appendix C, Eq. A20)

$$\beta = \frac{A_f}{A_i} \exp\frac{E_0^* - E^*(J_S^{out})}{k_b T}, \qquad (3)$$

where $A_i$ and $A_f$ are the HD areas before and after the change of $J_S^{out}$, respectively.

Generally, in accord with the qualitative consideration above, an increase in the spontaneous curvature of the proximal monolayer, $J_S^{out}$, in the range between $-0.1$ and $-0.065$ nm$^{-1}$, which corresponds to the variation of LPC area fraction in the proximal monolayer, $\phi$, between 0 and 10%, lowered the energy barrier for fusion pore formation and, hence, increased the acceleration factor, $\beta$ (Figs. 2b, c, d and 3). At the same time, we found the extent of this effect to be sensitive to the specific values of the system parameters, many of which remain to be reliably characterized.

According to our computations, the acceleration factor increases stronger for shorter intermembrane distances, $x^*$ (Fig. 2b,c). For $\kappa_m = 10$k$_B$T, $l = 1.5$ nm, and the area fraction of LPC in the proximal monolayer of about 10% ($\phi = 0.1$), the $\beta$ value changes from 2 to 8 when $x^*$ decreases from 8.6 nm to 4.6 nm. For large $x^*$ the fusion pore acceleration becomes negligibly small.

Another parameter modulating the acceleration factor, $\beta$, is the radius of the fusion site, $R_B$ (Fig. 2d,e). The larger the fusion site, the stronger the effect of $J_S^{out}$ variations on the acceleration factor. For a small fusion site of $R_B = 10$ nm and intermembrane distance $x^*$ of 8.6 nm, the stresses accumulated in the HD are large already in the initial state, before the variation of the proximal monolayer spontaneous curvature $J_S^{out}$. Therefore, the additional stresses generated by an increase in $J_S^{out}$ through, e.g., LPC addition, result in only small changes in the junction angle $\phi_0$, and the diaphragm radius, $R_D$, and thus lead to a very weak acceleration of the fusion pore formation (Fig. 2e). For a large fusion site with radius $R_B = 20$ nm an addition of 10% LPC to the proximal monolayers has a large effect characterized by the acceleration factor of $\beta = 50$ (Fig. 2e, green vs. red curves). For completeness, we considered also an alternative scenario in which the fusion site radius, $R_B$, is not restricted, but instead the distance between the HD rim and the fusion site boundary, $R_B - R_D$, is fixed. This scenario corresponds to the situation were the ring formed by the fusion proteins does not have a fixed radius and the number of fusion proteins creating this ring can change. We presented and discussed the results in Appendix D.

The acceleration factor depends on the background spontaneous curvature, $J_S^0$ (Fig. 3a, b). The splay in the HD rim is mostly negative for all monolayers. Therefore, an increase in $J_S^0$ results in high stresses already in the initial state so that the relative rise of the stress resulting from an increase in the proximal monolayer spontaneous curvature, $J_S^{out}$, is small. As a result, while for $J_S^0 = -0.1$ nm$^{-1}$ an addition of 10% LPC to the proximal monolayers results in the acceleration factor $\beta = 4$, for a more positive background spontaneous curvature of $J_S^0 = -0.02$ nm$^{-1}$ the same LPC addition is predicted to produce a negligibly small or even negative effect on the acceleration of the fusion pore formation.

The elastic energy and the stress-related phenomena can be reduced by the lipid redistribution between the bulk of the

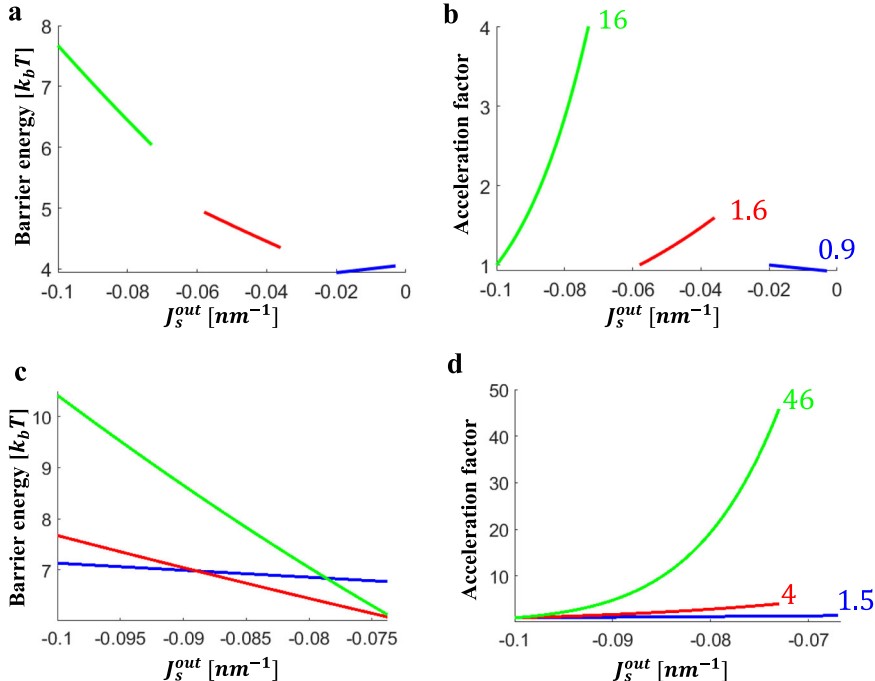

**Fig. 3 Dependence of the energy barrier and the acceleration factor of the fusion pore formation on the spontaneous curvature of the proximal monolayers and on the tilt decay length. a**, **b** describe the effects of varying background spontaneous curvature, $J_S^0$ on the energy barrier (**a**) and the acceleration factor (**b**): Blue line $J_S^0 = -0.02\,\text{nm}^{-1}$, red line $J_S^0 = -0.058\,\text{nm}^{-1}$, and green line $J_S^0 = -0.1\,\text{nm}^{-1}$. **c**, **d** describe the effect of tilt decay length, $l$ on the energy barrier (**c**) and the acceleration factor (**d**): Blue line $l = 1$ nm, red line $l = 1.5$ nm, and green line $l = 2$ nm. The parameter values used in all panels: $x^* = 6.6$ nm, $R_B = 15$ nm, $\kappa_m = 10 k_b T$, $\bar{\kappa}_m = -5 k_b T$, and $\lambda = 10$ pN. The tilt decay length size used in **a** and in **b** is $l = 1.5$ nm and the background spontaneous curvature in panels **c** and **d** is $J_S^0 = -0.1\,\text{nm}^{-1}$. In panels **b** and **d** the acceleration factor is indicated for the maximal calculated area fraction of LPC in the proximal monolayers. **b**: Blue line, $J_S^0 = -0.02\,\text{nm}^{-1}$, maximal fraction of LPC is 6%, $\beta = 0.9$. Red line, $J_S^0 = -0.06\,\text{nm}^{-1}$, maximal fraction of LPC is 6.8%, $\beta = 1.6$. Green line, $J_S^0 = -0.1\,\text{nm}^{-1}$, maximal fraction of LPC is 7.5% LPC, $\beta = 4$. **d**: Blue line, $l = 1$ nm, maximal fraction of LPC is 10%, $\beta = 1.5$ Red line, $l = 1.5$ nm, maximal fraction of LPC is 7.5%, $\beta = 4$. Green line, $l = 2$ nm, maximal fraction of LPC is 7.5%, $\beta = 46$.

membrane and the elastically stressed regions of the membrane monolayers. For instance, one may expect the depletion from the rim region of LPC molecules characterized by a large positive molecular curvature. We estimated the effects of lipid redistribution on the computed acceleration factor (Appendix E) and found that it lowers both the local LPC concentration in the rim region and the acceleration factor by about 20%. Such correction does not change the qualitative conclusions of our work.

Similarly, one could expect a fast flip-flop-driven redistribution between the membrane monolayers of such molecules as cholesterol to relax the elastic stress and, hence, affect the acceleration factor. However, it has to be appreciated that the monolayer area involved in the formation of the fusion site, which includes HD and its rim, is small compared to the area of the surrounding membrane. This means that the monolayers of the plasma membrane play a role of lipid reservoirs for the corresponding monolayers of the fusion site and set the compositions of these monolayers. The lipid flip-flop in the fusion site, if happening, is expected to be buffered by the molecule exchange with the reservoirs and, therefore, to have no effect.

Finally, the effect of the $J_S^{out}$ variations on the acceleration factor substantially depends on the elastic parameters of the system such as the lipid monolayer tilt, $\kappa_t$, and the saddle splay, $\bar{\kappa}_m$, moduli. The predicted sensitivity of the acceleration factor to the tilt modulus, $\kappa_t$, is substantial as presented in Fig. 3c, d, since $\kappa_t$ sets the value of junction angle in the HD rim, $\varphi_0$, and, hence, determines the total extent of the tilt deformation. The effects of the saddle splay modulus, $\bar{\kappa}_m$, are less substantial as evaluated in (Appendix F). We also found the acceleration factor to strongly increase for higher pore line tension, $\lambda$, (Appendix F).

In brief, our modeling supports the hypothesis that shifting the spontaneous curvature of the proximal monolayers towards more positive values generates additional stresses in the HD and accelerates opening of fusion pores.

**Experimental verification of the predictions of the model.** Our theoretical analyses demonstrated the feasibility of the mechanism in which shifting the spontaneous curvature of the proximal monolayers of the hemifused membranes to more positive values promotes the transition from hemifusion to complete fusion by accelerating fusion pore formation. To test this mechanism and to find out whether fusion dependence on Myomerger can be explained by its positive spontaneous curvature, we compared the effects of Myomerger on fusion with those of LPC.

We first tested whether LPC application to hemifused cells promotes their complete fusion in an assay based on cell fusion mediated by a well characterized fusogen, influenza HA[36–38]. In this assay, HA0-expressing cells (HA⁻cells) are treated with trypsin to convert fusion-incompetent HA0 into HA. Then these cells are brought into contact with red blood cells (RBC) labeled with both lipid and content probes. Fusion between HA-cells and bound RBC detected as redistribution of lipid and content probes from RBC to HA-cell is initiated by application of acidic pH medium that triggers fusogenic restructuring of HA. As reported earlier[5,38], at suboptimal trypsin concentrations, pH and temperature (room temperature vs. 37 °C), a considerable percentage of HA-cell/RBC pairs show lipid mixing but not content mixing, a hallmark of hemifusion phenotype. In this experimental model, expression of Myomerger[5] or application of sMyomerger[26-84] (Figs. S1 and S2) promote the transition from

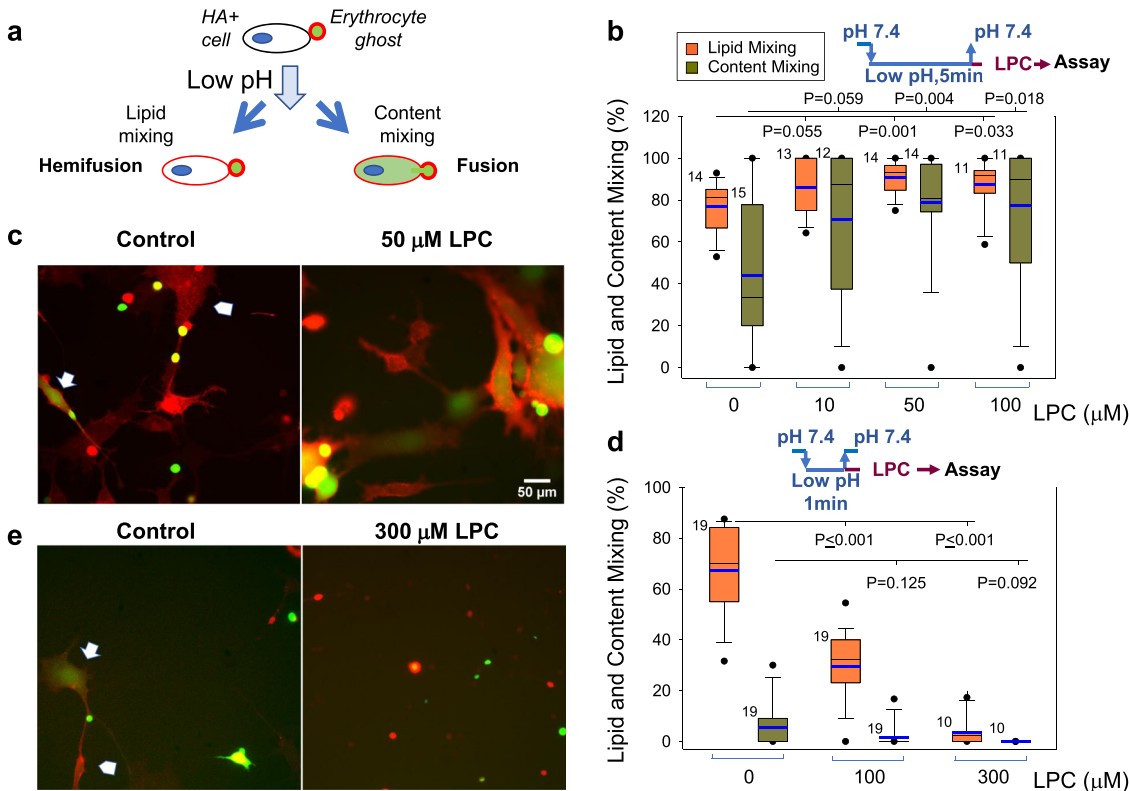

**Fig. 4 Dependent on the time of application and concentration, LPC either promotes hemifusion-to-fusion transition or inhibits hemifusion stage in HA-mediated cell fusion. a** Cartoon depicting influenza hemagglutinin (HA)-mediated fusion assay. HA-expressing 3T3 fibroblasts with pre-bound erythrocytes that were labeled with both lipid and content probes were treated with a low pH pulse. Lipid mixing without content mixing and lipid- and content- mixing report hemifusion and fusion, respectively. **b, c** HA-cells with pre-bound erythrocytes were treated with a 5-min pulse of pH 5.0 medium at 22 °C. Immediately after the end of low pH pulse, we applied the PBS supplemented with different concentrations of LPC. **d, e** HA-cells with pre-bound erythrocytes were treated with a 1-min pulse of pH 5.0 medium at 22 °C. Immediately after the end of low pH pulse, we applied the PBS supplemented with different concentrations of lysophosphatidylcholine (LPC) at the room temperature. **b, d** Fusion was analyzed with fluorescence microscopy 60 min after the low pH application. Content mixing (fusion) and lipid mixing (fusion plus hemifusion) were quantified as the ratios of the numbers of either content probe (carboxyfluorescein)-labeled HA-cells or lipid probe (PKH26)-labeled HA-cells, respectively, to the total number of HA-cells. Both **b** and **d** present representative results in one of three independent cell preparations each. *N*, the number of randomly selected fields of view examined for each condition is indicated to the left of the box associated with that data. Box-and-whisker plots show median (center line), mean (blue line), 25–75th percentiles (box), 10–90th percentiles (whiskers), and 5–95th percentiles (solid circles). *P*-values were calculated using a two-tailed unpaired *t*-test. **c, e** Representative images for the experiments in B taken for control cells and for the cells treated with 50 μM LPC (**c**) and for the experiments in **d** taken for the control cells and for the cells treated with 300 μM LPC (**e**). Arrows mark examples of fusion: HA-cells that acquired from fused RBC both lipid (red)- and content (green) probes. Arrowheads mark examples of hemifusion: HA-cells that acquired from fused RBC only lipid (red) probes. Scale bar 50 μm.

hemifusion to complete fusion. We now tested whether LPC has similar effects on fusion and found that application of neutral pH medium supplemented with LPC 5 min after a low pH application increased the number of completely fused cells (Fig. 4a–c). Thus, LPC application, as expression or application of Myomerger, shifts the observed fusion phenotype from hemifusion to full fusion.

As mentioned above, adding LPC to proximal leaflets of the membranes before their hemifusion inhibits hemifusion and fusion of the membranes in diverse fusion processes[2,4]. Indeed, when we brought the time of application of LPC closer to the end of the low pH treatment that initiated HA restructuring and hemifusion by applying the neutral pH medium supplemented with LPC 1 min rather than 5 min after the low pH application 300 μM LPC strongly inhibited lipid mixing (Fig. 4d,e). In similar experiments, 10 μM of sMyomerger[26-84] also strongly inhibited lipid mixing (Fig. 5a). As in the case of LPC, the 10 μM concentration of the peptide that strongly inhibited hemifusion (Fig. 5) was higher than the 2.5 μM concentration sufficient to promote fusion (Fig. S1). A small but significant increase in lipid mixing extent at 2.5 μM of sMyomerger[26-84] (Fig. 5a) can point to

formation of small and, possibly, short-living fusion pores that facilitate exchange of membrane probe but do not allow sufficient redistribution of a content probe to detect it in HA-cells. A retardation of the aqueous probe redistribution relative to membrane probe redistribution for small fusion pores in HA-cell/RBC fusion has been documented in Zimmerberg et al.[39].

Our findings suggested that both LPC and Myomerger promote fusion, if applied to already hemifused membranes, and suppress formation of hemifusion intermediates, if present in higher concentrations at the onset of fusion process. Indeed, while application of high concentrations of sMyomerger[26-84] to HA-cell/RBC complexes before application of low pH pulse inhibited both lipid and content mixing, low concentration of the peptide promoted complete fusion (Fig. S2). This result is consistent with our conclusions that (i) at high concentrations Myomerger inhibits HA-mediated hemifusion and complete fusion and (ii) there is a range of concentrations in which Myomerger does not block hemifusion and promotes fusion pore opening.

To compare the effects of sMyomerger[26-84] and LPC in the context of myoblast fusion, we used Myomerger-deficient

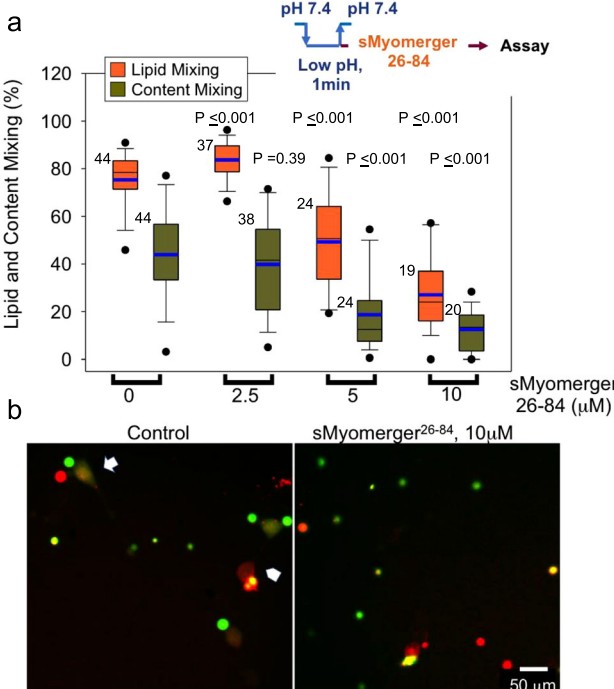

**Fig. 5 High concentrations of sMyomerger[26-84] inhibit HA-mediated hemifusion and fusion. a** Hemagglutinin (HA)-cells with pre-bound RBCs were treated with a 1-min pulse of pH 5.0 medium at the room temperature. Immediately after the end of low pH pulse, we applied PBS supplemented with different concentrations of sMyomerger[26-84] at the room temperature. Fusion (redistribution of lipid and content probes from RBCs to HA-cells) was analyzed with fluorescence microscopy. Content mixing (fusion) and lipid mixing (fusion plus hemifusion) were quantified as the ratios of the numbers of either content probe (carboxyfluorescein)-labeled HA-cells or lipid probe (PKH26)-labeled HA-cells, respectively, to the total number of HA-cells. N, the number of randomly chosen fields of view examined for each condition in one representative experiment out of three independent cell preparations, is indicated to the left of the box associated with that data. Box-and-whisker plots show median (center line), mean (blue line), 25–75th percentiles (box), 10–90th percentiles (whiskers), and 5–95th percentiles (solid circles). P-values were calculated using a two-tailed unpaired t-test. **b** Representative images for the experiments in **a** taken for the control cells and for the cells treated with 10 μM sMyomerger[26-84]. Arrow and arrowhead mark examples of fusion and hemifusion, respectively. Scale bar 50 μm.

C2C12 cells. A short-term application of sMyomerger[26-84] partially rescued fusion of these cells (Fig. 6b, c). LPC also promoted fusion of Myomerger-deficient cells (Fig. 6d, e), suggesting that hemifused intermediates that develop but do not progress in the absence of Myomerger, advance to expanding fusion pores in the presence of either sMyomerger[26-84] or LPC. Importantly, no fusion promotion was observed for concentrations of LPC (Fig. 6d), exceeding the 150 μM concentration used to inhibit both hemifusion and fusion in fusion synchronization experiments[5]. Higher concentrations of sMyomerger[26-84] also did not promote fusion (Fig. 6b). Thus, sMyomerger[26-84] and LPC similarly influence the fusion of Myomerger-deficient C2C12 cells, namely, promote fusion but only at moderate concentrations (Fig. 6). The lack of the promotion for higher concentrations of LPC and sMyomerger[26-84] can be explained if hemifusion connections that did not advance to fusion pores dissociate. If hemifusion connections appear and disappear continuously, the high concentrations of LPC and

sMyomerger[26-84] are expected to block appearance of new hemifusion connections but facilitate fusion for the cells that are already hemifused at the time of LPC or sMyomerger[26-84] application. In contrast, application of LPC or sMyomerger[26-84] concentrations that are low enough to allow hemifusion will, in addition, promote to fusion all new hemifusion connections developing during 30–60 min in the presence of the reagents.

Can hemifusion connections in myoblast fusion dissociate? The standard assay to detect hemifusion (lipid mixing without content mixing) does not distinguish situations where hemifusion structures are present at the time of analysis from situations where these structures had already dissociated by the time of the analysis but the cells had already exchanged lipid probes. Currently present hemifusion connections can be detected by an approach validated in HA-mediated fusion and other fusion processes[5,38,40,41]. In this approach, chlorpromazine (CPZ) added to the proximal leaflets of the membranes partitions into their distal leaflets, destabilizes hemifusion structures formed by these leaflets and reveals hemifusion by its conversion into complete fusion, detected as content mixing (Fig. 7a). To characterize the lifetime of hemifusion connections in Myomerger-deficient C2C12 cells, we used LPC block to accumulate ready-to-fuse differentiating cells upstream of hemifusion for 16 h[4,5]. LPC removal is expected to temporarily increase the numbers of hemifused cells. To evaluate the number of the cells connected by hemifusion structures at different times after LPC removal, we treated the cells with PBS supplemented with 50 μM CPZ for 1-min, and 60 min later scored CPZ-revealed hemifusion as complete fusion. We found that by the time LPC is removed by 3 washes with LPC-free differentiation medium (DM) hemifusion extents already reach maximum and, thus, ready-to-fuse myoblasts hemifuse within 1–2 min after LPC removal. The percentage of hemifused cells decreases two-fold from ~8% immediately after LPC removal to a background level of ~4% 30 min later (Fig. 7). This finding confirms the transiency of hemifused connections in myoblast fusion and suggests that differentiated Myomerger-deficient myoblasts maintain a background level of hemifusion by constantly forming transient hemifusion connections. While more accurate estimates of the hemifusion extents at different time points depend on yet unclear efficiency with which CPZ converts hemifusion into detectable full fusion, these experiments suggest ~15 min as a characteristic lifetime of hemifusion connections.

While application of moderate concentrations of sMyomerger[26-84] (or LPC) to Myomerger-deficient C2C12 cells significantly increased the fusion extents, these extents did not exceed ~4% (Fig. 6b–e). We reasoned that the observed fusion extents are limited by the numbers of the transiently hemifused cell pairs that happen to be present at the time of a short-term application of sMyomerger[26-84] and could be driven to fusion completion. Indeed, raising the numbers of the hemifused cell pairs present at the time of sMyomerger[26-84] application by the synchronization of the fusion process increased the levels of fusion observed after the peptide application to ~13% (Fig. 8b, c). Note that even without sMyomerger[26-84] application, the levels of fusion of Myomerger-deficient cells in the LPC-synchronization experiments were higher than those observed in the unsynchronized fusion of these cells. This finding suggested that LPC application converts the existing hemifusion connections into complete fusion and prevents formation of new hemifusion intermediates during 16-h incubation of the cells in the presence of LPC. This interpretation has been supported by finding that shortening the time interval between co-plating of the cells and LPC application expected to lower the numbers of hemifused Myomerger-deficient cells lowered the extents of fusion observed after LPC removal (Fig. S3).

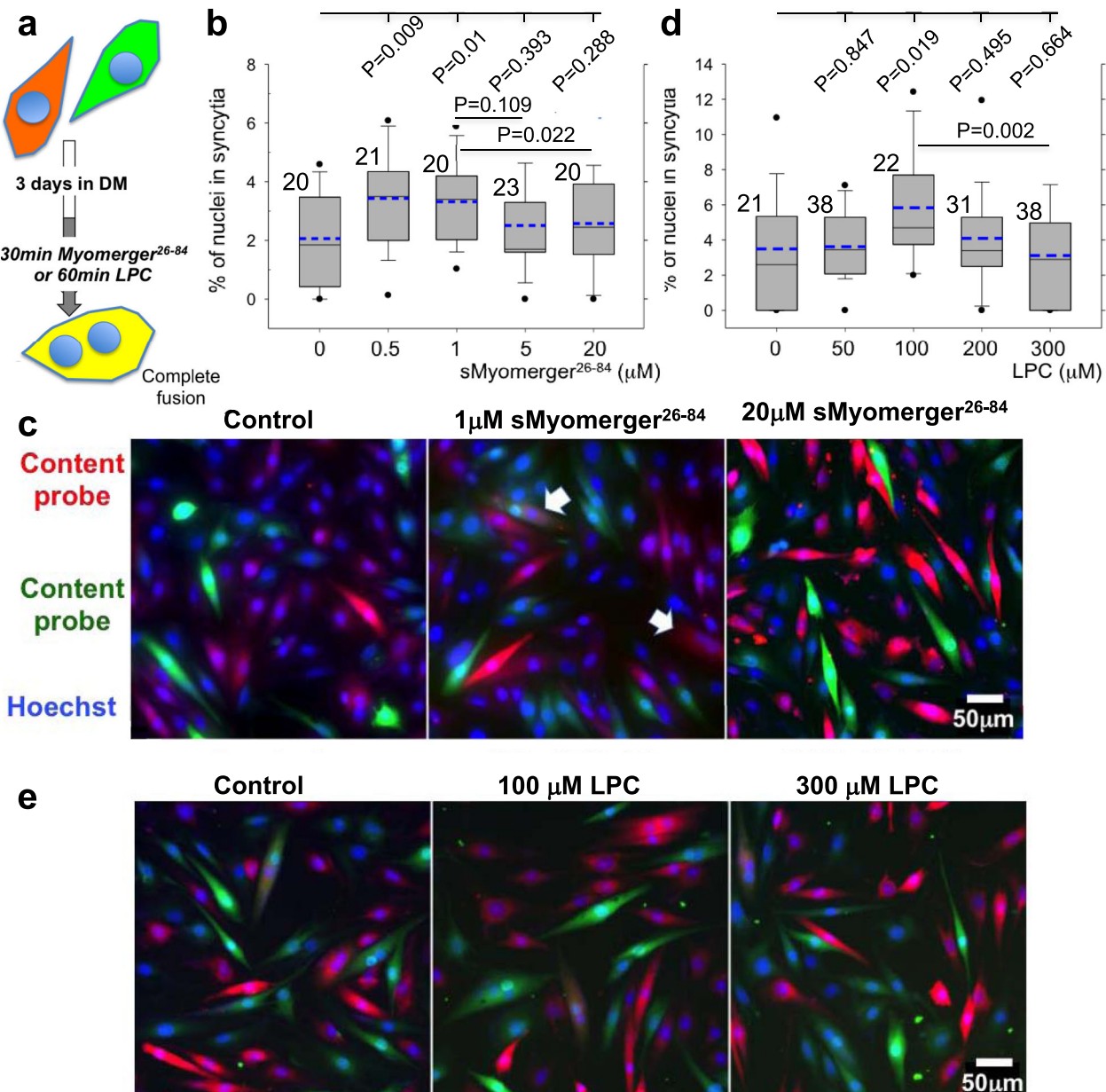

**Fig. 6 A short-term application of sMyomerger[26-84] or LPC to Myomerger-deficient myoblasts promotes their complete fusion. a** Schematic showing myoblast fusion assay. Myomerger-deficient C2C12 labeled with either orange cell tracker or green cell tracker were co-plated in the DM. Cell fusion was observed as formation of yellow (shown) or red or green multinucleated cells. After 3 days of the differentiation, we applied sMyomerger[26-84] for 30 min (**b**, **c**) or LPC for 60 min (**d**, **e**). Then we fixed the cells and scored fusion as the ratio of nuclei in the cells with ≥2 nuclei to the total number of nuclei. **b**, **d** show the quantifications of the fusion extents observed after application of sMyomerger[26-84] (**b**) or LPC (**d**). *N*, the number of randomly selected fields of view examined over three independent cell preparations for each condition, is indicated to the left of the box associated with that data. Box-and-whisker plots show median (center line), mean (blue line), 25–75th percentiles (box), 10–90th percentiles (whiskers), and 5–95th percentiles (solid circles). *P*-values were calculated using a two-tailed unpaired *t*-test. **c**, **e** Representative images for the experiments in **b** and **d** taken for the control cells and for the cells treated with 1 or 20 μM sMyomerger[26-84] (**c**) or 100 or 300 μM LPC (**e**). **c**, **e** Arrows mark examples of fusion. Scale bar 50 μm.

High enough spontaneous curvature of the proximal lipid monolayers prior to hemifusion is known to be detrimental for hemifusion and fusion[2]. Our findings above indicated that sMyomerger[26-84] generates positive spontaneous curvature of lipid monolayer and in higher concentrations suppresses hemifusion. Based on these findings, we suggested that while at physiological levels of expression in w.t. C2C12 cells Myomerger promotes fusion and, more specifically, post-hemifusion opening of a fusion pore[5], further raising the number of Myomerger molecules at the cell surface by adding sMyomerger[26-84] may

inhibit fusion. Experiments presented in Fig. 8d, e confirmed this prediction by demonstrating that at high enough concentrations of sMyomerger[26-84], fusion of w.t. C2C12 cells was considerably suppressed. This finding suggested that raising surface density of full-length Myomerger beyond the endogenous level characteristic for differentiating w.t. C2C12 cells will inhibit rather than promote fusion of these cells. While expression of Myomerger in Myomerger-deficient myoblasts rescues the fusion defect in these cells[11], a robust overexpression of this protein in w.t. C2C12 cells, confirmed by a considerable increase of its content in the cell

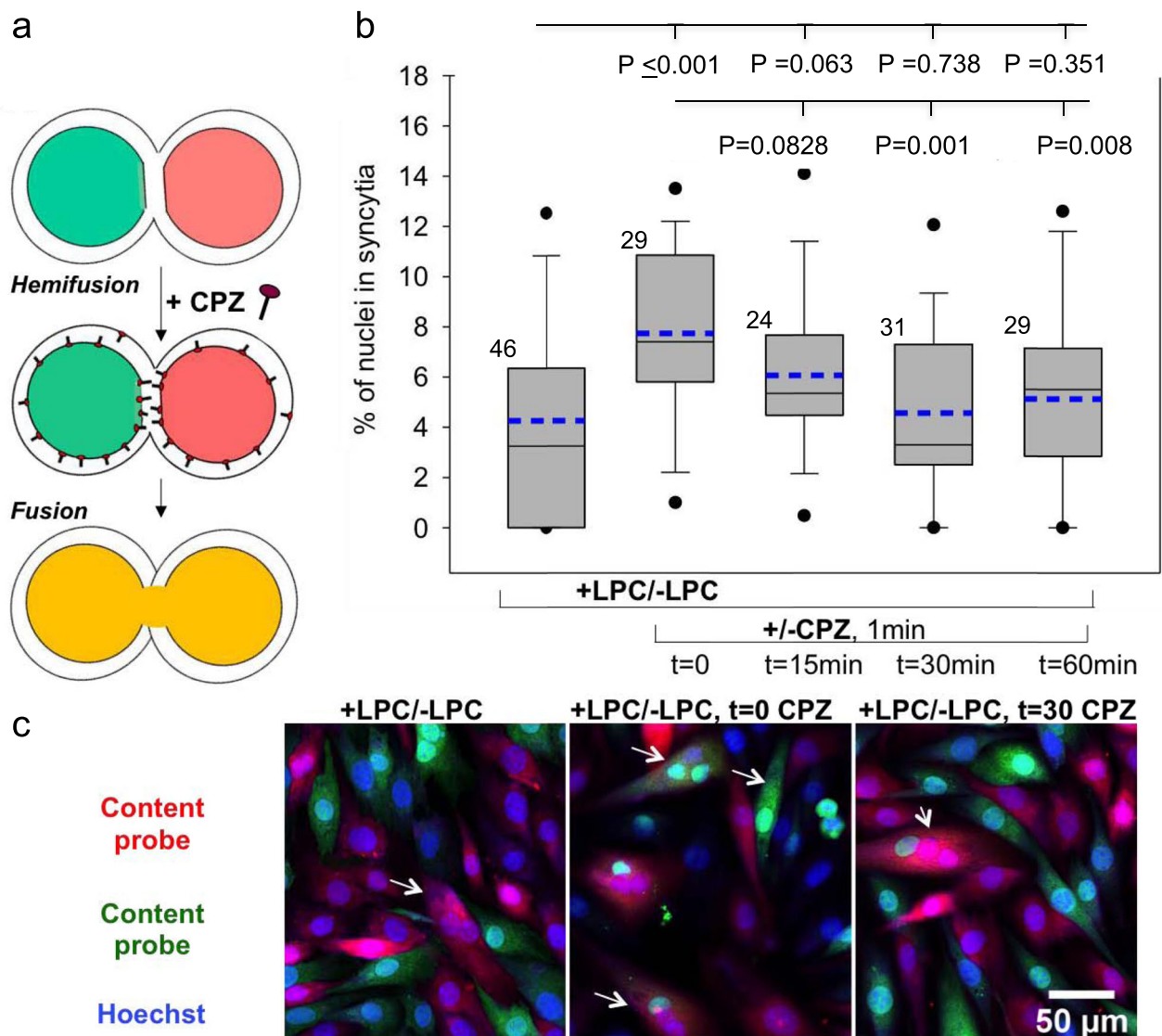

**Fig. 7 Time course of dissociation of hemifusion connections between Myomerger-deficient C2C12 cells. a** Chlorpromazine (CPZ) applied to the cells partitions into the inner leaflets of the plasma membrane, concentrates in the hemifusion structures composed of these leaflets, and converts hemifusion to full fusion. **b** Differentiating Myomerger-deficient cells were accumulated upstream of hemifusion by the incubation in the presence of lysophosphatidylcholine (LPC). After 3 days in the differentiation medium (DM), including last 16 h in the LPC-supplemented DM, we replaced this medium with LPC-free DM ($t = 0$) and, at different times after LPC removal, applied CPZ or not applied CPZ at all (bar to the left). At $t = 90$ min, we fixed the cells and scored fusion as the ratio of nuclei in the cells with ≥2 nuclei to the total number of nuclei. **b** N, the number of randomly selected fields of view examined over two independent cell preparations for each condition, is indicated to the left of the box associated with that data. Box-and-whisker plots show as median (center line), mean (dashed blue line), 25th–75th percentiles (box), 10–90th percentiles (whiskers), and 5–95th percentiles (solid circles). P-values were calculated using a two-tailed unpaired $t$-test. **c** Representative images for the experiments in **b** taken for the cells not treated with CPZ or treated at $t = 0$ or 30 min. Arrows mark examples of fusion. Scale bar 50 μm.

lysate, did not appreciably increase Myomerger content on the plasma membrane, evaluated by the surface biotinylation analysis (Fig. S4). While this hinders exploration of the effects of raising membrane concentration of full-length Myomerger beyond its endogenous level, our finding suggests the existence of a yet-unexplored mechanism that controls trafficking of Myomerger to/from plasma membrane.

The hypothesis that Myomerger drives hemifusion-to-fusion transition by generating the positive spontaneous curvature of lipid monolayer has been supported by finding that fusion of Myomerger-deficient cells can be also rescued by positive curvature-generating lipid LPC (see above) and peptide magainin 2[5]. We found that another positive curvature-generating peptide melittin[42], but not a negative curvature-

generating peptide penetratin[42], also promotes fusion pore opening in Myomerger-deficient C2C12 cells (Fig. 9).

We have also explored the effects of a negative curvature lipid oleic acid (OA)[2] and found that this lipid, in contrast to LPC, did not promote fusion pore formation in Myomerger-deficient cells (Figs. 9 and S5). OA application to HA-cell/ RBC pairs was reported earlier to promote hemifusion but inhibit the transition from hemifusion to fusion in HA-mediated fusion[38]. Similarly, while OA application to Myomerger-deficient cells did not promote formation of multinucleated cells, it promoted lipid mixing between the cells, as expected for a reagent that promotes hemifusion but not hemifusion-to-fusion transition (Fig. S5a, b). Furthermore, OA application to hemifusing LPC-synchronized w.t. C2C12

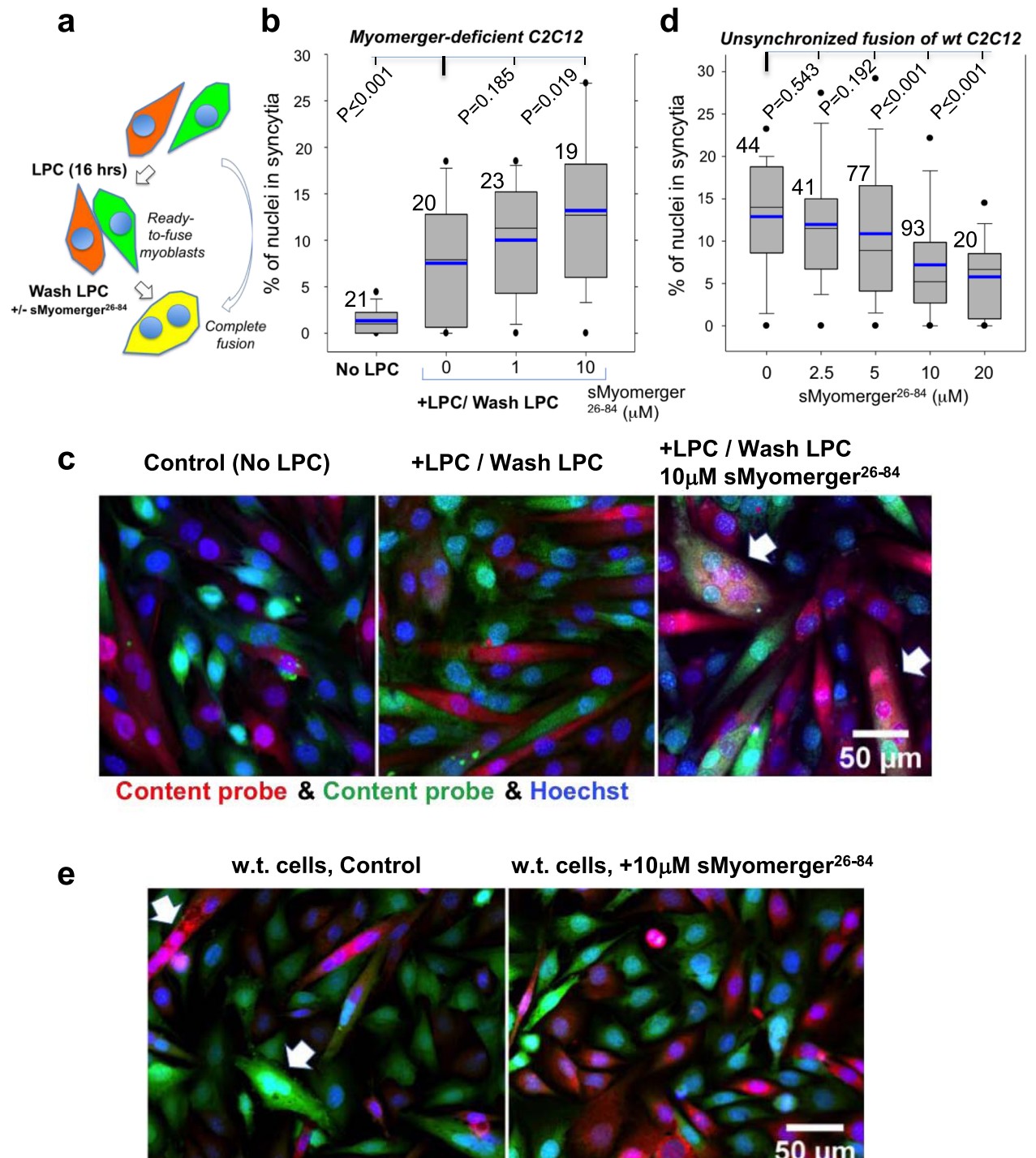

To estimate membrane concentrations of sMyomerger[26-84] that promote the hemifusion-to-fusion transition in Myomerger-deficient C2C12 cells, we treated the cells with tetramethylrhodamine labeled peptide (TAMRA-sMyomerger[26-84]) for 10 min and then washed to remove unbound peptide. As seen in Fig. 10a, similar to untagged peptide, TAMRA-sMyomerger[26-84] in moderate but not high concentration promotes opening of a fusion pore. We compared plasma membrane-associated fluorescence at fusion-promoting concentrations of TAMRA-sMyomerger[26-84] with fluorescence associated with membrane of giant lipid vesicles containing a known fraction of Lissamine Rhodamine B-tagged lipid. Based on this comparison, we roughly estimated the fusion-promoting membrane concentration of TAMRA-sMyomerger[26-84]

cells 5 min after LPC removal inhibited rather than promoted myoblast fusion (Fig. S5c, d). All these effects of a negative-curvature lipid OA are consistent with our hypothesis that positive curvature of LPC is essential for its ability to promote the hemifusion-to-fusion transition.

To summarize, our data on different curvature-generating peptides and lipids indicate that diverse ways of generation of positive but not negative curvature of proximal leaflets promote opening of fusion pores. Note, however, that in the case of OA we cannot exclude possible contribution of OA in distal leaflets, since OA added to the proximal monolayers of the membranes, in contrast to LPC, quickly redistributes between membrane monolayers.

**Fig. 8 Effects of LPC and sMyomerger[26-84] on synchronized fusion of Myomerger-deficient C2C12 cells and sMyomerger[26-84] on unsynchronized fusion of wild type (w.t.) C2C12 cells. a** Within 3 days period of differentiation, Myomerger-deficient C2C12 cells were labeled with either orange cell tracker or green cell tracker, and then treated with the hemifusion inhibitor lysophosphatidylcholine (LPC) in the differentiation medium (DM) for 16 h to accumulate myoblasts that are ready to fuse. LPC removal ('LPC wash') allowed the cells to undergo hemifusion and complete fusion. sMyomerger[26-84] in different concentrations was applied for 1 h at the time of LPC removal. In another experimental design, Myomerger-deficient C2C12 cells differentiated for 3 days (**b, c**) or w.t C2C12 cells differentiated for 2 days (**d, e**) were allowed to fuse without LPC block. **b** Quantification of synchronized fusion in the presence of different LPC concentrations and fusion extents observed at the same time post-differentiation for unsynchronized fusion (No LPC) for Myomerger-deficient C2C12 cells. **c** Representative images for the experiments in **b** taken for the No LPC, 0 and 10 µM sMyomerger[26-84] experiments. Arrows mark examples of fusion. Scale bar 50 µm. **d** W.t. C2C12 cells differentiated for 2 days, labeled with either orange cell tracker or green cell tracker and co-plated in the DM in the presence of different concentrations of sMyomerger[26-84]. Fusion was scored 4 h after co-plating. **e** Representative images for the experiments in D taken for the 0 and 10 µM sMyomerger[26-84] experiments. Arrows mark examples of fusion. Scale bar 50 µm. **b, d** Fusion was quantified by assessing formation of multinucleated cells as the ratio of nuclei in the cells with $\geq 2$ nuclei to the total number of nuclei. N, the number of randomly selected fields of view examined over three independent cell preparations for each condition, is indicated to the left of the box associated with that data. Box-and-whisker plots show median (center line), mean (blue line), 25–75th percentiles (box), 10–90th percentiles (whiskers), and 5–95th percentiles (solid circles). P-values were calculated using a two-tailed unpaired t-test.

as ~470 Myomerger ectodomain peptide molecules/µm² of the plasma membrane.

In summary, our experimental findings support the proposed mechanism in which Myomerger promotes opening of fusion pores in myoblast fusion by generating positive contributions to the spontaneous curvature of the proximal monolayers of the fusing plasma membranes.

## Discussion

Myomaker[10] and Myomerger[11–13], two muscle-specific proteins involved in myoblast fusion, control distinct stages of the fusion reaction with Myomerger responsible for the transition from the Myomaker-dependent hemifusion to the stage of a fusion pore formation[5]. In this study, we propose a physical mechanism in which interactions of the Myomerger ectodomain with proximal (outer) monolayers of the hemifused membranes promote formation of a fusion pore by generating a positive contribution to the monolayer's spontaneous curvature. Specifically, the variations of the proximal monolayer's spontaneous curvature, $J_S^{out}$, generate elastic stresses in the region of the three-membrane junction that forms the rim of the HD. These stresses propagate into the HD membrane and promote the fusion pore formation there, hence, accelerating the hemi-to-full fusion transition. We confirmed the feasibility of this mechanism by theoretical analysis, in which we computationally explored the dependence of the rate of the pore formation in the HD on the spontaneous curvature of the merged proximal monolayers, $J_S^{out}$. As a specific way to modify $J_S^{out}$, we considered supplementing the proximal monolayers with a lipid of a positive effective molecular curvature, LPC.

We also verified the key assumptions of the proposed mechanism by the experiments in which we found the Myomerger ectodomain sufficient to generate positive contributions to the monolayer's spontaneous curvature. An observed similarity between the Myomerger- and LPC- effects on fusion of Myomerger-deficient myoblasts and on HA-mediated cell fusion further substantiated the mechanism. While our experiments have been focused on a function of a muscle-specific protein Myomerger, the mechanistic insights into this poorly characterized stage of hemifusion to fusion pore transition may be relevant to other protein-mediated membrane fusion processes. An effective and fast transformation of the hemifusion intermediate into the fusion pore is critically important for the efficiency of the fusion reaction because the intermembrane hemifusion connections that do not convert to the fusion pores either dissociate letting the membranes separate[41], or remain long-living and do not give rise to fusion pores, thus representing a branch off the productive fusion pathway[38,43]. Furthermore, a slow advance along the fusion pathway can allow for a non-productive dissipation of the energy released by the restructuring of fusogenic proteins[21,41]. Because of the expected shortness of the lifetime of hemifusion intermediates that can give rise to fusion pores, even a relatively modest acceleration of the pore opening can be critically important for the fusion efficiency.

**Fusion mechanism.** A number of earlier studies have analyzed different aspects of membrane fusion focusing on the conformations of the proteins involved[44] or lipid rearrangements in fusion by computer simulations[45–47] and by modeling the membranes as macroscopic continuous films[48,49]. Our study utilizes the latter approach and focuses on the hemifusion to fusion pore transition. Our model assumes that Myomerger molecules accelerate the fusion pore formation remotely, acting from outside of the HD through generation of the elastic stresses in the membrane leaflets rather than serving as structural components of the fusion pore. The hypothesis that proteins catalyze a fusion pathway by modulating the intra-membrane elastic stresses is supported by the similarities between the intermediate structures and lipid dependencies reported for different biological fusion processes, including cell–cell fusions such as myoblast fusion, and for fusion of protein-free bilayers[1,4,5,50]. Our analysis indicates that while a substantial positive contribution to the spontaneous curvature of the proximal monolayers, $J_S^{out}$, of the membranes is detrimental for hemifusion, a moderate increase in $J_S^{out}$ promotes conversion of the hemifusion intermediates into fusion pores and completion of the fusion process. Our modeling indicates that the effects of $J_S^{out}$ on the rate of the fusion pore formation depend on the parameters used in the analysis (e.g., the distances between the bilayers, the radius of the fusion site, the background spontaneous curvature of the monolayers, the pore line tension, the decay length of the tilt deformations and the saddle splay modulus).

The configuration of the fusion site at the time and place of cell–cell fusion in muscles and even in the much better characterized fusion processes of viral entry and $Ca^{2+}$ triggered exocytosis remains to be fully characterized. The range of the radii of the fusion site used in our analysis is based on the reports that the radii of the tight contact regions formed by the SNARE complexes and by HIV Env can be estimated as ~10 nm[51] and ~20 nm[52], respectively. In our modeling, we varied the distances between neutral planes of the opposing proximal monolayers, at the boundary of the fusion site in the 4.6–8.6-nm range. Considering that the neutral planes underlie the ~1nm-thick monolayer polar head regions, this range corresponds to ~2.6–6.6 nm water gaps between the two membranes. These

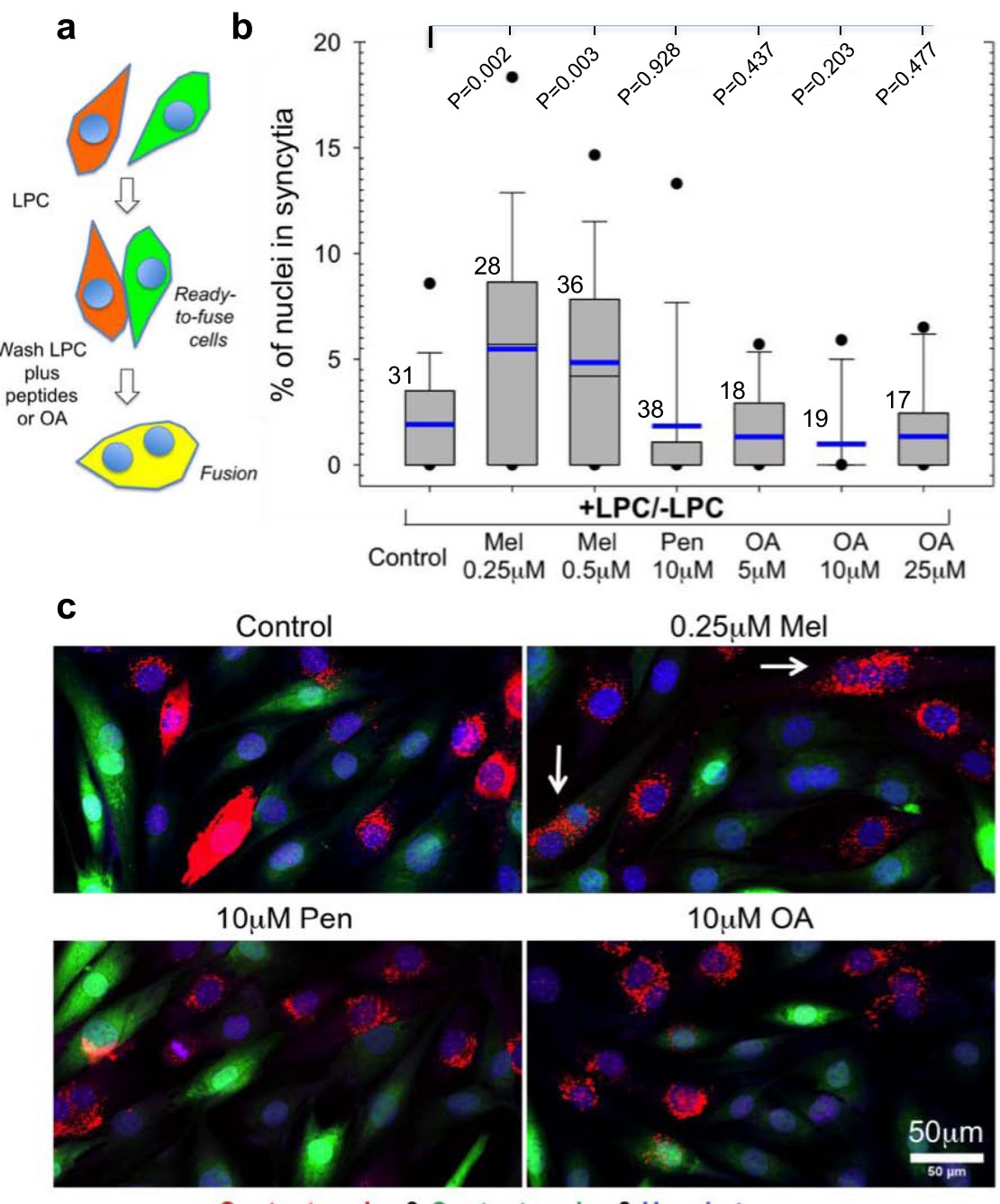

**Fig. 9 A short-term application of melittin but neither penetratin nor oleic acid to Myomerger-deficient myoblasts synchronized by LPC promotes complete fusion of myoblasts. a** Schematic showing synchronized myoblast fusion assay. **b** Differentiating Myomerger-deficient cells were accumulated upstream of hemifusion by the incubation in the presence of lysophosphatidylcholine (LPC). After 3 days in the differentiation medium (DM), LPC-supplemented DM was replaced with LPC-free DM supplemented or not with melittin (Mel) or penetratin (Pen) or oleic acid (OA). 60 min later we fixed the cells and scored fusion as the ratio of nuclei in the cells with ≥2 nuclei to the total number of nuclei. *N*, the number of randomly selected fields of view examined for each condition over two independent cell preparations, is indicated to the left of the box associated with that data. Box-and-whisker plots show median (center line), mean (blue line), 25–75th percentiles (box), 10–90th percentiles (whiskers), and 5–95th percentiles (solid circles). *P*-values were calculated using a two-tailed unpaired *t*-test. **c** Representative images for the experiments in **b** taken for the control cells and for the cells treated with 0.25 μM Mel or 10 μM Pen or 10 μM OA. Arrows mark examples of fusion. Scale bar 50 μm.

gaps are close to the characteristic 2.5–5 nm diameters of the ectodomains of the post-fusion hairpin conformations of several well-studied viral fusogens and the SNARE complex rod, which pull the membranes together and remain laying between the fusing membranes[21,22]. Based on the cryo-electron tomography analysis, the gaps between the membranes in HA-mediated fusion can be as narrow as ~2 nm[53]. Therefore, considering that the shorter the intermembrane distance, the stronger the acceleration of the pore formation (Fig. 2), the distance of 6.6 nm used in most of our analysis, is a conservative estimate. The same is true for our choice of the value of the pore line tension of $\lambda = 10$ pN, which corresponds to DOPC membranes. Indeed, as

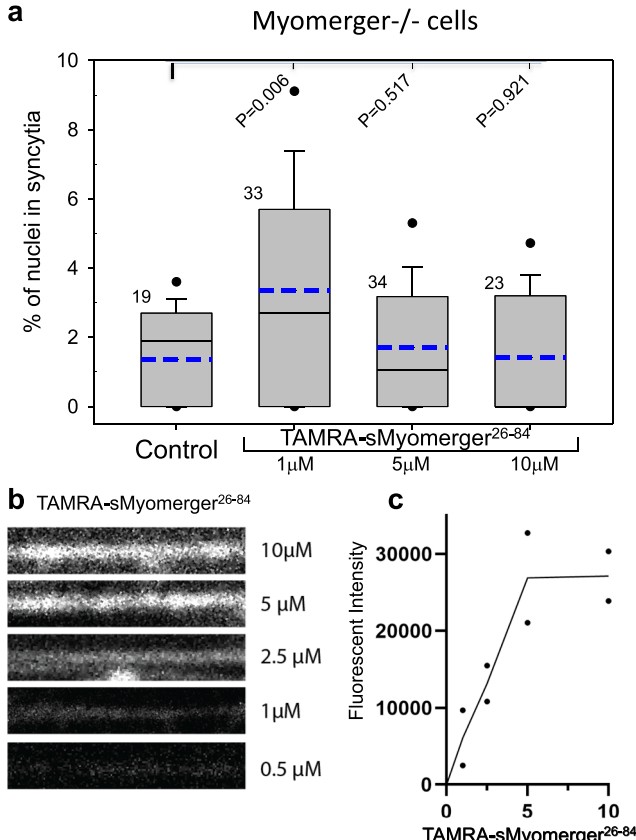

**Fig. 10 Membrane concentration of sMyomerger[26-84] that promotes the hemifusion-to-fusion transition in Myomerger-deficient C2C12 cells.**
**a**, **b** After 3 days in the differentiation medium (DM) unlabeled Myomerger-deficient C2C12 cells were washed and incubated for 10 min in the DM with different concentrations of TAMRA-sMyomerger[26-84]. Then the cells were thoroughly washed to remove unbound peptide, and either fixed for analysis of plasma membrane-associated TAMRA fluorescence (**b**) or incubated in the DM for 50 more minutes and fixed for fusion quantification (**a**). **a** Fusion was quantified by assessing formation of multinucleated cells as the ratio of nuclei in the cells with ≥2 nuclei to the total number of nuclei. N, the number of randomly selected fields of view examined for each condition over two independent cell preparations, is indicated to the left of the box associated with that data. Box-and-whisker plots show median (center line), mean (blue line), 25–75th percentiles (box), 10–90th percentiles (whiskers), and 5–95th percentiles (solid circles). P-values were calculated using a two-tailed unpaired t-test. **b** The representative images of membrane-associated TAMRA labeling for the cells treated with different concentrations of TAMRA-sMyomerger[26-84]. **c** Quantification of the intensity of membrane-associated TAMRA fluorescence in two independent experiments shown as points with line connecting the means for each of the peptide concentrations used.

seen in Fig. A5, acceleration factor of the fusion pore formation is expected to be much higher for larger values of $\lambda$ such as $\lambda = 27\,pN$ reported in Portet and Dimova[35].

The spontaneous curvature of the plasma membrane is conferred by its composition and our estimate of the spontaneous curvature of the distal (inner) monolayers and of the background spontaneous curvature of the proximal (outer) monolayers is based on the values of the effective molecular curvature of the abundant lipid, PC. Experimental confirmation of the key results of the analysis, including the conclusion that the lipid of a positive molecular curvature, LPC, promotes the hemifusion-to-fusion transition, indirectly validates the ranges of this parameter used in the modeling.

Better experimental characterization of the model parameters will further test the validity of our model in its application to different membrane fusion processes.

**Myoblast fusion.** Myomerger expression in myoblasts is induced during myogenesis and correlates with the development of the fusogenic potential of myoblasts[11]. Our data indicate that the ectodomains of Myomerger anchored in the outer monolayer of the plasma membrane shift the spontaneous curvature of this monolayer to more positive values. Dependent on the level of the Myomerger expression at the cell membrane, this shift can promote fusion by accelerating the hemifusion-to-pore transition or, at significantly higher membrane concentration, inhibit fusion by suppressing hemifusion. Our finding that overexpression of full-length Myomerger does not increase its concentration at the plasma membrane (Fig. S4) suggests that due to regulatory mechanisms controlling its plasma membrane localization, in physiological conditions, Myomerger never reaches hemifusion-inhibiting concentrations.

Our estimate of the surface density of sMyomerger[26-84], application of which to Myomerger-deficient cells allows Myomerger to promote the overall fusion reaction in formation of multinucleated myotubes, ~500 molecules/µm² of the plasma membrane is in the range characteristic for viral fusogens (~2500 HA trimers/µm² in HA-cells[37] and ~400 HIV Env trimers/µm² [54]). Furthermore, our finding that the fusion defect in Myomerger-deficient myoblasts can be partially rescued not only by application of sMyomerger[26-84] but also by application of LPC substantiates the earlier conclusion that fusion promotion by Myomerger does not depend on any specific interactions between Myomerger and other proteins at the surface of the fusion-committed cells[5].

While Myomerger ectodomain partially rescues fusion of Myomerger-deficient myoblasts, the level of fusion remains much lower than that observed for w.t. myoblasts. Our results suggest that sMyomerger[26-84] promotes unsynchronized myoblast fusion only if present at the stage where these cells establish transient hemifusion connections. For the synchronized cells, application of LPC and Myomerger ectodomain brings the efficiency of fusion of Myomerger-deficient myoblasts to levels approaching those observed for w.t. C2C12 cells (~15% in the synchronized fusion assay in Fig. 8b in Myomerger-deficient myoblasts and in Fig. S5 in w.t. myoblasts). In addition to the transiency of hemifusion intermediates, the efficiency of fusion promotion by Myomerger ectodomain can be limited by a limited lifetime of the Myomerger ectodomain at the cell surface. The full-sized protein is anchored in the membrane by its transmembrane domain and, perhaps, involved in the interactions with other membrane components. These interactions with membrane and protein partners may provide additional levels of control of the Myomerger-dependent myoblast fusion stages.

It remains to be seen whether the suggested mechanism, in which Myomerger accelerates the post-hemifusion stages of the myoblast fusion pathway through generating positive contributions to the spontaneous curvature of the proximal membrane leaflets, is utilized by proteins involved in other biological fusion processes. Most of the suggested mechanisms of protein-mediated fusion have explained only how fusion proteins drive hemifusion by bringing proximal monolayers into a very close partially dehydrated contact[14,15] and by localized membrane dimpling (reviewed in Chernomordik et al.[2]). The factors promoting pore formation and expansion in the HD, the most energy-demanding stages of the fusion (reviewed in Chernomordik et al.[2]), were either not explicitly discussed or associated with specific properties of the transmembrane domains of the fusion proteins and

their interactions with other regions of the proteins. Since replacing transmembrane domains of SNAREs and HA with a lipid anchor did not prevent fusion pore opening[36,55–57], we consider this scenario unlikely at least for some fusion processes. The transition from hemifusion-to-fusion was also suggested to be driven by fusion protein-independent forces, such as the global membrane tension[58]. Our analysis suggests an alternative mechanism for this transition. Many of the proteins that mediate viral and intracellular fusion have functionally important membrane-interacting amphiphilic regions[59–62] that are likely to modify the spontaneous curvatures of the membrane leaflets. Other mechanisms of promotion of fusion pore formation by membrane-interacting regions of viral fusogens were modeled by molecular simulations of pore-stalk complexes[45,46]. Future work will show whether the mechanism of fusion pore formation by induction of positive contributions to the spontaneous curvature of the proximal membrane monolayers, proposed here for Myomerger-dependent myoblast fusion, could be also utilized in other fusion processes.

In conclusion, our study demonstrates that the ability of Myomerger to shift the spontaneous curvature of the outer leaflets of the myoblast membranes towards positive values plays a critical role in the Myomerger's function in myoblast fusion. Understanding of this feature of the fusion mechanism may inform novel methods of treating muscle diseases where muscle fusion is impacted. It will be interesting if a similar physical mechanism underlies poorly understood transition from early fusion intermediates to fusion completion in viral fusion, in exocytosis and, perhaps, in the transition from hemi-fission to fission in endocytosis[63–66].

## Methods

**Cells**. C2C12 cells were purchased from American Type Culture Collection. Myomerger$^{-/-}$ C2C12 cells generated previously using CRISPR/Cas9 mutagenesis[11] were propagated in DMEM-Glutamax (Gibco) containing 10% fetal bovine serum (FBS, GIBCO Life technologies # 10437-028) and supplemented with penicillin-streptomycin (1%) at 37 °C and 5% CO$_2$. The cells were committed to myogenic differentiation by switching to DMEM containing 5% horse serum (HS, ThermoFisher, Catalogue # 26050088) and antibiotics (referred to as differentiation medium (DM)). NIH 3T3 mouse fibroblasts of clone 15 cell line that stably express HA of Japan strain of influenza were a kind gift of Dr. Joshua Zimmerberg, NICHD, NIH[67]. These HA-expressing cells were cultured in DMEM with 10% FBS, 10$^4$ units/mL penicillin G, and 10 mg/mL streptomycin at 37 °C and 5% CO$_2$. Human red blood cells (RBCs) were isolated from the blood of healthy donors (who had consented to participate in the NIH IRB-approved Research Donor Program in Bethesda, MD; all samples were anonymized).

**Reagents**. A 10-mg/ml stock solution of lauroyl LPC (Avanti Polar Lipids, catalogue # 855475 P) was freshly prepared in water. Synthetic Myomerger$^{26-84}$ and TAMRA-sMyomerger$^{26-84}$ synthesized by LifeTein, NJ, USA were added to the cells from a 1 mM stock solution in PBS. To explore the effects of short-term (≤1 h) applications of sMyomerger$^{26-84}$ or LPC to the cells, the cell medium bathing the attached cells was replaced by 1 ml of medium supplemented with either sMyomerger$^{26-84}$ or LPC in the stated final concentrations. Fluorescent lipid Vybrant DiI, membrane-permeant Green CMFDA cell tracker and orange CMRA Cell Tracker were purchased from ThermoFisher Scientific (#V22885; #C7025 and #C34551, respectively). Penetratin and melittin were purchased from AnaSpec (catalogue # AS-64885) and Sigma-Aldrich (catalogue # M2272), respectively, and used in concentrations 0.25 and 0.5 μM for melittin; and 5 and 10 μM for penetratin, based on earlier studies[68] and[69], respectively. Melittin and penetratin in concentrations exceeding 1 μM and 10 μM, respectively, caused apparent cytotoxicity. Oleic acid (OA) was purchased from Sigma (catalogue #O1008). Chlorpromazine (CPZ) (Sigma) was prepared as a 0.5-mM solution in PBS. At the time of LPC removal and at different times after it, synchronized Myomerger−/− C2C12 cells were placed into PBS supplemented with 50 μM CPZ for 60 s. Then the medium was replaced with the DM for 60 min at 37 °C.

**Preparation of the myoblasts for fusion and synchronized fusion assays**. We labeled w.t. and Myomerger-deficient C2C12 cells 2 days after placement in the DM with either membrane-permeant Green CMFDA cell tracker or orange CMRA Cell Tracker[4,70] (Fig. S6a, b, c). At this time, we already see elongated myoblasts and, for w.t. cells, some myotubes. We gently lifted differently labeled cells with

0.05% trypsin-EDTA (Thermo Fisher, catalog number: 25300054), mixed the cell suspensions 1-to-1 and vortexed them to randomize distribution of differently labeled cells. Then we plated the cells in the DM on collagen-coated dishes. Probes were used to facilitate identification of fused cells. After one more day of differentiation, i.e., ~3 days after placing the cells into the DM (Fig. S6a), we fixed the cells with Formalin 10% buffered in Phosphate for 10 min at the room temperature. In a modification of this assay (the experiments presented in Fig. S5a), we detected early fusion intermediates that allowed lipid mixing but did not advance to formation of multinucleated cells by co-plating the cells labeled with green cell tracker and cells labeled with fluorescent lipid DiI, as described in[70]. Hemifused cells were identified as mononucleated cells labeled with both of our probes. Note that by the time of analysis DiI is already partially internalized[70].

In some experiments (Fig. S6c), we synchronized myoblast fusion and uncoupled it from myogenic differentiation using reversible fusion inhibitor LPC[4,71]. In brief, 2–3 h after co-plating differently labeled cells, we placed fusion-committed myoblasts into 2 ml of the complete DM supplemented with 150 μM lauroyl LPC. 16 h later, we washed the cells three times with the LPC-free DM within 1–2 min. If not stated otherwise, we fixed the cells, as described above, one hour after LPC removal, at ~3 days post-differentiation.

In the experiments presented in Fig. 8d,e, we shortened the differentiation period (~2 days instead of ~3 days in all other experiments, Fig. S6a–c) to apply the sMyomerger$^{26-84}$ at the time of the most robust fusion, when changes in the fusion efficiency have stronger effects on measured fusion extents. In this experimental design, the cells were fixed, as described above, at ~2 days post-differentiation.

**Myoblast fusion quantification**. After fixing the cells, we stained the nuclei with Hoechst (10 mg/ml stock diluted 1,000-fold for 30 min at the room temperature) and took the images on fluorescence microscope using appropriate excitation and emission filters and Micro-Manager v.1.4.23 Open Source Microscopy Software for image acquisition. We also used Fiji/ImageJ open-source image processing package v.2.1.0/1.53c for viewing and scoring. Fusion completion was detected as formation of multinucleated cells (defined as cells with more than 1 nucleus per cell) and quantified as the percentage of cell nuclei present in multinucleated cells normalized to the total number of cell nuclei. For each condition, ≥10 randomly chosen fields of view were imaged. Note that lifting C2C12 cells for labeling after 2 days of differentiation resulted in a loss of many already formed myotubes[72], explaining why fusion index in our experiments was considerably lower than that in publications, in which myoblasts were not labeled.

Since at the time we scored fusion, differentiated C2C12 already do not divide, we counted all cells with more than 1 nucleus as fused. In several experiments, we verified that counting as fused cells only double labeled cells plus cells that are not double labeled but have more than 2 nuclei does not change the observed effects, including myoblast fusion dependence on the time interval between co-plating and LPC application (compare Fig. S3b, d) and a gradual loss of hemifusion with time after LPC removal (compare Fig. 7b and Fig. S7).

**Liposome preparation**. All lipids (dioleoyl phosphatidylcholine, DOPC: dioleoyl phosphatidylserine, DOPS; TopFluorPE; RhPE, TopFluorPC, and TopFluo TMR PC) were purchased from Avanti Polar Lipids. Large unilamellar vesicles (LUVs) were prepared by extrusion. In brief, lipids dissolved in benzene/methanol (95:5, vol:vol) were freeze-dried under high vacuum overnight and the dried lipid powder was hydrated at room temperature in a buffer containing 100 mM NaCl, 10 mM Hepes, 5 mM EDTA, pH 7.0 and vortexed. The resulting lipid suspension was submitted to 10 successive cycles of freezing and thawing by successively immersing the vial containing the lipid suspension into liquid nitrogen and a warm water bath. Thereafter the lipid suspension was extruded 10 times through two stacked polycarbonate filters of 100 nm pore-size (Nucleopore, Whatman) using a LIPEX extruder (Northern Lipids Inc., Burnaby, Canada) to produce large unilamellar vesicles.

**FRET measurement**. Liposomes containing DOPC:DOPS:TopFluorPE:RhPE (89:10:0.5:0.5 mole percent) were used to measure changes in FRET efficiency for probes placed in the lipid headgroup region. Liposomes containing DOPC:DOPS: TopFluorPC:TopFluorTMRPC (89:10:0.5:0.5 mole percent) were used to measure changes in FRET efficiency for probes placed in the lipid tails region. Emission and absorption spectra of individual fluorescent lipids are presented in the Fig. S8. Emission spectra were measured in liposomes, containing 0.5 mole percent of the corresponding fluorescent lipid, 89.5 mole percent of DOPC and 10 mole percent DOPS. Absorption spectra were measured in methanol. We calculated R$_0$ values to be 57 Å and 58 Å for fluorophores placed in the headgroup region and lipid tails region, respectively, using Eq. 1 from Wu and Brand[73]. We used the following values for parameters: quantum efficiency of both donors in absence of acceptor = 0.94, peak extinction coefficient for RhPE = $9.6 \times 10^4$ M$^{-1}$cm$^{-1}$, peak extinction coefficient for TopFluorTMRPC = $5.9 \times 10^4$ M$^{-1}$cm$^{-1}$, orientation factor – 2/3, index of refraction – 1.4. Fluorescent emission spectra were recorded on a Fluoromax 4 spectrofluorometer (Horiba, USA) using FluoroEssence v.2.1 software package supplied by the instrument manufacturer. Excitation wavelength and bandwidth were set to 480 nm and 5 nm, respectively. Emission bandwidth was set to 2 nm, with a range of 490–650 nm with 1 nm step and integration time set to 0.1 s. Liposomes were added

into 2 mL of buffer in which liposomes were prepared to the final lipid concentration of 2.5 μM. Lipids or peptides were added to the cuvette in small increments and fluorescence spectra were recorded after each addition. We observed no photo-bleaching of any of our fluorophores in the conditions used. FRET efficiency was characterized as a change in quantum efficiency of donor emission due to the presence of acceptors $E = 1 - I_{DA}/I_D$, where $I_{DA}$ is fluorescence intensity of donor in presence of acceptor and $I_D$ is florescence intensity of donor in absence of acceptors or at infinite dilution of acceptor. We used RStudio v.1.2 free software package to estimate $E$ and $N$ for each condition using linear fit of the following equation:

$$S = N \times (1 - E) \times S_D + N \times K_1 \times S_A + N \times E \times K_2 \times S_A \qquad (4)$$

where $S$ is the measured spectra, $S_D$ is the spectra of donor alone, $S_A$ is the spectra of acceptor alone, $N$ is number of fluorophores, $K_1$ and $K_2$ are constants explained below. First element of the sum above corresponds to the emission of donor fluorophores. Second element of the sum corresponds to the emission of acceptor fluorophores due to direct excitation by excitation beam and constant $K_1$ adjusts for the differences in the extinction coefficient, quantum efficiency and number of fluorophores between donor and acceptor. Last element of the sum corresponds to the emission of the acceptor fluorophores due to energy transfer with constant $K_2$ adjusting difference in quantum efficiency of donor and acceptor. Spectra of donor ($S_D$) and acceptor ($S_A$) were measured using liposomes of the same composition but substituting the other fluorophore with DOPC. Constant $K_1$ was estimated from the spectra of liposomes containing both donor and acceptor fluorophores in presence of detergent (0.5% DDM). In this case $S = N \times S_D + N \times K_1 \times S_A$ and $K_1$ is equal to the ratio of $S_A$ and $S_D$ contribution. Constant $K_2$ was estimated from paired spectra of liposome containing both fluorophores before and after detergent addition. In this case $E$ can be estimated as the ratio of the $S_D$ contribution before detergent addition and after detergent addition and $K_2$ calculated from the linear decomposition of spectra before detergent addition using estimates for $K_1$ and $E$.

**Fusion experiments with HA-cells**. To examine the effects of sMyomerger[26-84] and LPC on fusion mediated by influenza hemagglutinin (HA), we used Japan HA-expressing NIH 3T3 mouse fibroblasts of clone 15 cell line (HA-cells)[67]. As described in Leikina et al.[5], HA-cells were cultured at 37 °C and 5% CO₂ in DMEM supplemented with 10% heat-inactivated FBS and antibiotics. HA-cells express a fusion-incompetent uncleaved precursor form of HA, HA0. We labeled RBCs with the fluorescent lipid PKH26 (Sigma) and loaded with carboxyfluorescein, CF (5-(and-6)-Carboxyfluorescein, mixed isomers, #C368, Invitrogen), as in Chernomordik et al.[38]. Before the experiments, we cleaved HA0 at the surface of HA-cells into the fusion-competent HA1-S-S-HA2 form (referred to as HA) with trypsin. As in Leikina et al.[5], to observe hemifusion phenotype (RBC-to-HA-cell redistribution of PKH26 without redistribution of CF) rather than only complete fusion (redistribution of both PKH26 and CF), we lowered the percentage of cleaved and, thus, fusion-competent HA by using suboptimal conditions for trypsin cleavage. We treated HA-cells with 1 μg/ml trypsin for 5 min at room temperature instead of optimized conditions (5 μg/ml, 10 min, room temperature) that yielded enough HAs to observe close to 100% complete fusion. HA-cells were twice washed with PBS and incubated for 10 min with a 1-ml suspension of RBCs (0.05% hematocrit). We washed HA-cells with PBS to remove unbound RBC. Then the cells were exposed to the low pH medium (PBS titrated with citrate). To increase the percentage of hemifused HA-cell/RBC pairs, we triggered fusion by treating the cells with pH 5.0 or pH 5.1 medium at the room temperature rather than with pH 4.9 medium at 37 °C[38]. At the end of low pH application, we replaced the low pH medium with PBS supplemented or not supplemented with LPC or sMyomerger[26-84]. In the experiments presented in the Fig. S2, sMyomerger[26-84] was applied before low pH pulse. HA-cells with pre-bound RBCs were incubated for 5 min with different concentrations of sMyomerger[26-84] then treated with a 3-min pulse of pH 5.0 medium at the room temperature. Immediately after the end of low pH pulse, we raised the temperature to 37 °C. Fusion (also referred to as complete fusion) and hemifusion between HA-cells and RBCs were assayed by fluorescence microscopy 1 h after low pH application. Fusion and hemifusion were quantified as the ratios of the numbers of either content probe (CF)-labeled HA-cells or lipid probe (PKH26)-labeled HA-cells, respectively, to the total number of HA-cells.

We verified that applications of low pH, sMyomerger[26-84] and LPC did not influence the number of HA-cell associated RBCs per field, defined as the sum of the number of unfused RBCs (RBCs bound but not fused to HA-cells detected as labeled RBC bound to unlabeled HA cell) and the number of fused/hemifused RBCs per field (scored as the number of HA-cells that acquired content- and/or membrane probes). Thus, changes in fusion/hemifusion efficiencies could not be explained by changes in the efficiency of HA-cell/RBC docking.

To verify that sMyomerger[26-84] enters the tight contact zones, we applied TAMRA-sMyomerger[26-84] peptide to HA-cell/RBC complexes. These cells are known to establish extended contact with an area on the order of tens of square microns that are characterized by a relatively constant intermembrane distance of ~13 nm[74]. Fluorescence labeling observed throughout the contact zone within 5 min after application of the peptide to pre-bound cells (Fig. S9) argued against hypothesis that sMyomerger[26-84] is excluded from the tight contact zone.

Fusion extents and, especially, fusion/hemifusion ratio in the suboptimal conditions are very sensitive to relatively minor changes in HA expression[38]. As a result, for different batches of cells and on different days, treating HA-cell/RBC

pairs with the same pulses of low pH (for instance, 1 min application of pH 5 at the room temperature) resulted in different efficiencies of hemifusion and fusion. Since fusion extents varied from day to day, we routinely started the experiments by choosing the precise conditions of the low pH treatment. Each experiment presented here was repeated at least three times, and all functional dependencies reported were observed in each experiment. The data presented were averaged from the same set of experiments. In all cases, all the data presented in one panel of the figure for different conditions have been gathered in the parallel experiments carried out at the same time.

**Evaluating cell surface density of sMyomerger[26-84] at the plasma membrane**. Myomerger-deficient C2C12 cells incubated in the DM for 3 days without labeling were washed thrice in PBS and one time in the DM at 37 °C to remove debris. One hour later we applied different concentrations of TAMRA-sMyomerger[26-84] in 1 mL. After 10 min incubation, we washed the cells with PBS 3 times to remove the unbound peptide and one time with the DM and fixed the cells either right away or after 50 more minutes in the DM. These cells were then analyzed for cell fusion quantification, as described above, and for the confocal microscopy analysis of the amount of membrane-associated peptide, respectively.

To estimate the surface density of the bound peptide, in two independent experiments, we selected representative regions of plasma membranes and quantified TAMRA fluorescence for each concentration of TAMRA-sMyomerger[26-84]. To estimate the surface density of the bound peptide at the cell membrane, we used as a fluorescence intensity standard giant unilamellar liposomes formed, as described in[75], from a 9:1 mix of DOPC and DOPS doped with 0.5 mol % of Liss Rhod PE (1,2-dioleoyl-sn-glycero-3-phosphoethanolamine-N-(lissamine rhodamine B sulfonyl, Avanti Polar Lipids, catalogue # 810150). Based on a known amount of RhDOPE in liposomes and taking 0.7 nm² as an area per polar head of lipid, we calculated the surface density of fluorophores in the GUV membrane. Then we compared the fluorescence intensity of the peptide signal at the cell surface with the fluorescence intensity of the liposome membrane and, thus, estimated the surface density of the peptide. Our estimate assumes that liposome and cell membrane have similar geometry and there is one fluorophore per peptide. We corrected for the differences in the spectral properties of dyes using absorption and emission spectra and quantum efficiencies reported for unconjugated dyes (we used quantum efficiency of Alexa 555 for TAMRA, since they are reported by the manufacturer to be similar and we did not find the value for TAMRA).

**Surface biotinylation analysis of the cells overexpressing Myomerger**. Platinum E cells (PE, Cells Biolabs) were plated on 100 mm culture plates at confluency of $5 \times 10^6$ cells per plate 24 h before transfection to generate retrovirus. Ten micrograms of the retroviral plasmid DNA (pBabe Empty or pBAbe Myomerger) were transfected into PE cells using FuGENE 6 (Promega). In all, 48 h after transfection, the viral media were collected, filtered through a 0.45-μm cellulose syringe filter and mixed with polybrene (Sigma) at a final concentration of 6 μg ml⁻¹. The C2C12 cells were plated on 100 mm culture plates at a density of $4 \times 10^5$ cells per plate 16–18 h before infection and were infected with 9 mL of the viral media. After 24 h of infection the viral media was removed, washed with PBS, and cells were trypsinized and propagated for further analysis.

Two 150 mm plates of C2C12 cells (transduced with empty vector or overexpressing Myomerger) were differentiated for two days and washed three times with cold PBS (137 mM NaCl, 2.7 mM KCl, 10 mM Na₂HPO₄, KH₂PO₄ pH 8.0). Cells were treated with 5 mL of 0.5 mg/mL EZ-link Sulfo-NHS-SS-Biotin (Pierce) at room temperature with gentle rocking for 30 min. A control plate not exposed to Biotin was also processed. The cells were washed once with PBS and twice with 50 mM Tris-HCl, pH 7.5 to block non-reacted biotin. The cells were scraped with 2 mL of lysis buffer (10 mM Tris-HCl pH 7.5, 150 mM NaCl, 0.1% Triton X-100, 100 μM oxidized glutathione (Sigma-Aldrich), Protease Cocktail (Roche)) and were lysed with sonication five times on ice. Samples were centrifuged at a speed of $21,230 \times g$ for 20 min and supernatants were incubated with 40 μL of pre-washed magnetic streptavidin beads (Pierce) at room temperature for one hour to isolate biotinylated proteins. Beads were washed at least six times with the lysis buffer. For immunoblots, 30 μL of RIPA buffer (50 mM Tris-HCl pH 7.4, 1% Triton X-100, 1% sodium deoxycholate, 1 mM EDTA, 0.1% SDS) was added along with 8 μL of 5X Laemmli loading buffer (300 mM Tris-HCl pH 6.8, 10% SDS, 50% glycerol, 0.25% bromophenol blue) to the beads and heated at 95 °C for 5 min. Western blotting was performed for both Myomerger (R & D systems, 1: 200) and GAPDH (Millipore, 1:10000).

**Theoretical modeling**. Simulations with the developed models have been carried out using MATLAB R2018b and Surface Evolver 2.70, as detailed in the Appendixes.

**Quantification and statistical analysis**. If not stated otherwise, the data from multiple fields of view examined over three independent experiments were analyzed using a parametric two-tailed $t$-test to determine significance. Data are presented in box-and-whisker plots where the center lines show the median, box the 25th-75th percentiles, whiskers above and below the box indicate the 90th and 10th percentiles, solid circles show 5th/95th percentile. These statistical analyses

were performed using the SigmaPlot13 software. The criterion for statistical significance was $P \leq 0.05$ (*$P \leq 0.05$, **$P < 0.01$, ***$P < 0.001$).

## Data availability

Data supporting the findings of this manuscript are available from the corresponding authors upon reasonable request. A reporting summary for this Article is available as a Supplementary Information file.

## Code availability

All files used in the theoretical analysis are uploaded to Github account (https://github.com/GonenGonen/Myomerger-Hemifusion) and any further details are available from the corresponding authors upon reasonable request.

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

## Acknowledgements

We thank Dr. Joshua Zimmerberg, NICHD, NIH for the kind gift of hemagglutinin expressing cells. The research in L.V.C. laboratory was supported by the Intramural Research Program of the Eunice Kennedy Shriver National Institute of Child Health and Human Development, National Institutes of Health. M.M.K. is supported by SFB 958 "Scaffolding of Membranes" (Germany) and Singapore-Israel (NRF-ISF) research grant 3292/19 and holds Joseph Klafter Chair in Biophysics. This work was also supported by grants to D.P.M. from the Cincinnati Children's Hospital Research Foundation, National Institutes of Health (R01AR068286, R01AG059605), and Pew Charitable Trusts.

## Author contributions

G.G. and M.M.K. have carried out theoretical analysis. E.L. and L.V.C. designed cell fusion experiments; E.L. carried them out and analyzed the data. K.M. and L.V.C. designed FRET experiments; K.M. and G.L.-O. carried them out and K.M. analyzed the data. J.M.W. has carried out the analysis of the surface concentration of sMyomerger (Fig. 10). D.G.G. and D.P.M. have designed the surface biotinylation experiments (Fig. S4) and D.G.G. has carried them out and analyzed. D.P.M. contributed to the interpretations of the results. G.G., L.V.C., and M.M.K. wrote the manuscript with assistance from all authors.

## Competing interests

The authors declare no competing interests.
