## [Peer Review File · Nature Communications]

REVIEWER COMMENTS

Reviewer #1 (Remarks to the Author):

Golani and co-workers study myomerger, a membrane protein involved in the formation of multinucleated cells. In particular, this protein is important in the late stages of cell-cell fusion, from Hemifusion to pore-opening formation. The authors suggest a mechanism whereby myomerger changes the curvature of the outer membrane to more positive values; generating elastic stress (lateral tension) which propagates into the hemifusion complex. The mechanism is further validated with myoblast fusion and influenza hemagglutinin-mediated cell-cell fusion. The shift in spontaneous positive curvature is postulated as a general mechanism for other fusion processes in development and viral infections.

Overall this is very interesting study and reveals an important mechanism on the role of myomerger during the last steps of the fusion reaction. Moreover, this transition between hemifusion and pore opening formation could be extrapolated to other fusogens, and therefore this study is of high interest and broad scope.

The experiments were carefully designed and the stats are well employed in my opinion.

Lipids with an inverted-cone shape such as LPC arrest fusion at a hemifusion stage.

In myoblast fusion two proteins are key, myomaker (required for hemifusion) and myomerger (drives opening of a fusion pore).

The shift toward spontaneous curvature of proximal monolayers to positive values driven either by LPC or sMyomerger inhibits hemifusion and also full fusion. However, once hemifusion occurs both help to complete the fusion reaction, precisely by changing the curvature of the hemifusion diaphragm toward positive values. Please discuss these events for other fusion processes (virus-fusion) but also during endocytosis and exocytosis.

Fig 1

FRET experiments:

Amphipathic membrane insertions, which generate positive contributions in membrane curvature will increase the distance between donors and acceptors bound to the polar heads of lipids as compared to dyes placed at the end of lipid acyl chains

FRET pair lipid heads: TopFLuor PE (donor) / Rhod PE/ Please describe this FRET pair (Fluorescein – Rhodamine? Introduce the spectral properties and why it was chosen as a good FRET pair.

FRET pair acyl chain: (TopFLuor PC / TMR) Please justify the FRET couple as above. Explain in both cases how different photobleaching coming from the donor and/or the acceptor might affect your results.

-How did you measure the FRET efficiency. Do you mean apparent FRET efficiency, as authors utilized an intensity-based method?

When calculating the apparent FRET efficiencies in the presence/absence of Myomerger were these measurements carried out with low light levels to avoid photobleaching.

Do you think that the concentration (ratio of donor and acceptor fluorophores) played a role in your experiments? Did you try different concentrations of donor and acceptor to check for reproducibility?

Experimental verification of the predictions of the model:

Cell-Cell fusion experiments performed with HA-cells and bound RBC.

-Addition of LPC before hemifusion arrests fusion reaction whilst cells in which hemifusion occurred (different concentrations, lower) induce fusion pore formation.

High concentrations of Myomerger right after hemifusion will also suppress fusion.

To further validate the hypothesis of low concentrations of myomerger playing an important role to drive hemifusion to fusion in the cell-cell experiments one could think of an experiment in which myomerger is labelled (not sure if a fluorescent protein might kill function here) and once could assess the fusion process as a function of time (from hemifusion to fusion pore formation and stabilisation or flickering), see procedure in [Jones et al., 2017 Cell Rep]. If labelling of the myomerger is not possible (even in the intracellular domain), perhaps one could fix the cells after the cell-cell kinetic experiments and perform immunostaining. This way one might find a relationship between myomerger expression and fusion (in all steps).

Reviewer #2 (Remarks to the Author):

In this manuscript the authors provide a theoretical mechanism for the observation that a specific protein, myomerger, promotes fusion in myoblast cells.

The proposed mechanism assumes that myomerger proteins, which adsorb at the proximal leaflets of the fusing membranes, induce stress in the hemifusion diaphragm and thereby promoting opening of the full fusion pore.

The plausibility of the theory is, on the one hand, supported by calculations showing how reasonable choices for the parameters involved indeed lead to a speed-up in fusion rate, and, on the other hand, by supporting experiments showing that (i) indeed myomerger promotes positive curvature to a similar extent as lysolipids such as LPC, and (ii) addition of myomerger, as well as LPC, can rescue cells that are trapped in hemifused states.

Despite the overall elegance of the proposed theory and the apparent support from the experimental measurements, I am not convinced that we now understand the mechanism through which myomerger acts, for the following reasons:

- 1) First of all, the proposed mechanism seems to be very generic. According to the theoretical model, any membrane surface active compound (albeit peptide or protein, or anything that embeds at the membrane surface) would promote opening of the fusion pore. In the current study, this has been shown for LPC, but I would like to see more examples, and, in particular, are there negative controls?
- 2) It remains unclear how the myomerger proteins could adsorb in the vicinity of the HD to exert their curvature-induced stress, as the HD is surrounded by the proteins of the fusion machinery which keep the fusing membranes together.
- 3) The theoretical calculations depend on many parameters. As the authors acknowledge, most of these parameters are not well known and hard to measure in the first place, even for single component lipid systems. In vivo, we are dealing with a much more complicated situation with many different lipids and other proteins involved. For instance, the authors argue that "Since the spontaneous curvature of the distal monolayers remains unchanged, this can happen only at the expense of deviation of the distal monolayers from their optimal configurations within the junction region." However, a change in lipid composition could be another (and energetically less costly) way to induce changes without the need for elastic stresses building up in the HD. Such changes could occur on very fast time scales (requiring only local lipid redistributions). In addition, the role of lipid flipflops (notably cholesterol) should be discussed in

potentially affecting the release of stress.

4) It also remains puzzling what concentrations are needed to promote fusion. It is stated that LPC can promote fusion pore opening (from cells that are trapped in a hemifused state) most efficiently at 50 microM, and that 300 microM is needed to block hemifusion. But what happens to cells that are treated with 50 microM before hemifusion takes place? Is full fusion still observed or not? The same question arises wrt myomerger.

5) The explanation of the results from the cell experiments are difficult to interpret, which is not surprising given the complexity of cell fusion. The manuscript contains some speculative text on why certain experimental findings are not in line with the predictions from the theory (e.g., arguments about why synchronized cells show efficient fusion in absence of LPC or myomerger, arguments about the relevance of a 4% speedup, arguments about why high concentrations block fusion, etc.). The manuscript would be strengthened with more convincing explanations or additional experiments on these topics.

6) Even if we take for granted that addition of myomerger or LPC in hemifused cells indeed have a positive effect on fusion pore opening, to me it seems that ANY type of shock you add to the cell at this stage would have such an effect. After all, the hemifused state is likely a kinetically trapped state requiring some form of energetic input to be released. Are there any negative controls?

7) To bridge the gap between the theory on one side, and the full cell experiments on the other side, what I miss in this study are experiments performed on liposomal fusion assays. Here, many of the predictions from the theory could be much more straightforwardly tested.

8) At the end of the manuscript the potential role of fusion peptides is mentioned. According to the authors, such fusion peptides (which are also known to be embedded at the membrane surface) could also promote fusion pore opening via the same mechanism. In this light, simulation studies show indeed a mechanism by which fusion peptides can induce fusion pores, namely via promoting formation of stalk-pore complexes. It would be nice to cite these studies and discuss whether the creation of stalk-pore complexes could play a role here as well. See for instance work by Risselada and colleagues, and Fuhrmans & Marrink.

Reviewer #3 (Remarks to the Author):

In this manuscript, Golani et al. investigated how the fusogenic peptide myomerger facilitates the transition from hemifusion stalk to fusion pore formation during membrane fusion. They hypothesized that myomerger generates a positive spontaneous curvature of the proximal monolayer. The membrane elastic stresses then propagate into the hemifusion diaphragm, which accelerates fusion pore formation. Interestingly, the effect of myomerger ectodomain is strikingly similar to that of LPC, which is known to generate a positive spontaneous curvature of lipid monolayers, further validating the hypothesis.

This study aims to fill a major gap in our understanding of membrane fusion, namely how a hemifusion stalk resolves into a fusion pore. The theoretical modeling is effective in showing that positive curvature of the proximal monolayers generates additional stress in the hemifusion diaphragm

and accelerates fusion pore formation. Experiments in cells largely validated the model. Overall, this is a very interesting study that provides novel insights into a critical step during membrane fusion.

Major comments:

1. The authors showed that both LPC and sMyomerger (at low concentration) facilitate the transition from hemifusion stalk to fusion pore formation, likely by inducing positive curvature of the proximal monolayer. To support this conclusion, the authors should test whether a lipid that generates negative curvature would inhibit this transition.
2. If high concentration of sMyomerger inhibits the transition from hemifusion stalk to fusion pore formation, does overexpressing a high level of wild-type Myomerger in myomerger-deficient cells inhibit myoblast fusion?
3. Based on the previous study (Leikina et al., 2018), myomerger-deficient cells should be stalled at the hemifusion stage. After the incubation scheme in Figure 6, what was the percentage of myomerger-deficient cells that have formed hemifusion stalks? How long would the potential fusion partners stay attached by the hemifusion stalks?
4. After releasing the LPC block in Figure 7, did the percentage of cells linked by hemifusion stalks increase, by how much? Can all of these potential fusion partners linked by hemifusion stalks proceed with fusion in the presence of low concentrations of LPC and sMyomerger? How fast is each fusion event? Why would high concentrations of sMyomerger not inhibit fusion after releasing the LPC block?

Minor comments:

1. In the HA-RBC fusion experiments, lipid mixing (indicating hemifusion) seems to be much higher compare to a previous study (Leikina et al., 2018). How to explain the difference?
2. Does HA activation change the percentage of HA-cell/RBC pairs, which in turn could partially explain the increased number of full fusion events?
3. In Figure 4C, there are few bright green cells without red signal in the 50 μ M LPC-treated cells. What are these cells?
4. In Figure 4D, the control showed an average of 70% lipid mixing and less than 10% content mixing. With the same setting, Figure 5A showed similar lipid mixing, but significant more content mixing (>40% vs. 10% in Figure 4D). Given that these were similar experiments, it is difficult to appreciate the inhibitory effect driven by sMyomerger if the control content mixing can be as low as 10%.
5. In Figure 5A, 2.5 μ M sMyomerger seemed to significantly promote lipid mixing compared to the control, but did not enhance content mixing. Could the authors provide some explanations for this?
6. The statistical differences in Figure 6B and 6D are relatively small, even when the fusion index was counted as any cells with more than 1 nucleus. It is necessary to make sure by live imaging that the binucleate cells did not result from incomplete cytokinesis.
7. In Figure 7B, the myomerger-deficient cells were allowed to differentiate for 72 hrs before mixing, whereas the WT cells in Figure 7C only differentiated for 48 hrs and allowed to fuse for 4 hrs. Why did the authors use much longer differentiation time for the myomerger-deficient cells? Due to the different experimental conditions, it is difficult to compare the levels of fusion between WT cells and myomerger-deficient cells synchronized to fuse.

8. In the method section, the timelines for myoblast fusion are somewhat confusing. Why were the cells plated in differentiation medium for ~85 hours? Also, the fixation time should be indicated differently for different experiments (not ~ 85 hr post differentiation for all experiments), e.g. the WT C2C12 cells were only allowed to differentiate for 48 hrs compared to the 72 hrs differentiation time for the myomerger-deficient cells.

POINT-BY-POINT RESPONSE

We would like to thank you and all three Reviewers for the careful review of our manuscript and insightful and valuable comments and critiques.

The list of new experiments/figures added in Revision includes:

Fig. 7 shows that hemifusion connections in myoblast fusion are transient rather than irreversible;

Fig. 9 and Fig. S5 show that a positive-curvature generating peptide melittin, similarly to sMyomerger²⁶⁻⁸⁴ and magainin 2 and lipid lysophosphatidylcholine, and in contrast to, negative curvature-generating peptide penetratin and lipid oleic acid, rescues fusion of Myomerger-deficient C2C12 cells;

Fig. 10 shows estimation of membrane concentrations of the sMyomerger²⁶⁻⁸⁴ promoting hemifusion-to-fusion transition using fluorescent peptide;

Fig. S2 shows that high concentration of sMyomerger²⁶⁻⁸⁴ before onset of hemagglutinin-mediated fusion inhibits hemifusion and fusion, while low concentration of the peptide promotes complete fusion;

Fig. S3 explores the time course of development of hemifusion after co-plating differently labeled Myomerger-deficient cells;

Fig. S4 reports that a robust overexpression of Myomerger in w.t. C2C12 cells is not accompanied by appreciable increase of Myomerger on the plasma membrane suggesting the existence of a yet-unexplored mechanism that controls trafficking of Myomerger to/from plasma membrane;

Fig. S9 indicates that sMyomerger²⁶⁻⁸⁴ readily enters the tight contact zone.

Appendix E describes theoretical modelling of the effects of lipid redistribution.

Our reply to each of your comments below are in bold red font.

Reviewer #1 (Remarks to the Author):

Golani and co-workers study myomerger, a membrane protein involved in the formation of multinucleated cells. In particular, this protein is important in the late stages of cell-cell fusion, from Hemifusion to pore-opening formation. The authors suggest a mechanism whereby myomerger changes the curvature of the outer membrane to more positive values; generating elastic stress (lateral tension) which propagates into the hemifusion complex. The mechanism is further validated with myoblast fusion and influenza hemagglutinin-mediated cell-cell fusion. The shift in spontaneous positive curvature is postulated as a general mechanism for other fusion processes in development and viral infections.

Overall this is very interesting study and reveals an important mechanism on the role of myomerger during the last steps of the fusion reaction. Moreover, this transition between hemifusion and pore opening formation could be extrapolated to other fusogens, and therefore this study is of high interest and broad scope.

The experiments were carefully designed and the stats are well employed in my opinion.

Lipids with an inverted-cone shape such as LPC arrest fusion at a hemifusion stage.

In myoblast fusion two proteins are key, myomaker (required for hemifusion) and myomerger (drives opening of a fusion pore).

1) The shift toward spontaneous curvature of proximal monolayers to positive values driven either by LPC or sMyomerger inhibits hemifusion and also full fusion. However, once hemifusion occurs both help to complete the fusion reaction, precisely by changing the curvature of the hemifusion diaphragm toward positive values. Please discuss these events for other fusion processes (virus-fusion) but also during endocytosis and exocytosis.

In the revised MS we have added a following comment “It will be interesting if a similar physical mechanism underlies poorly understood transition from early fusion intermediates to fusion completion in viral fusion, in exocytosis and, perhaps, in the transition from hemi-fission to fission in endocytosis¹⁻⁴.” We also mention in the MS that “Many of the proteins that mediate viral and intracellular fusion have functionally important membrane-interacting amphiphilic regions that are likely to modify the spontaneous curvatures of the membrane leaflets.” We respectfully suggest that adding more specific discussion of possible contributions of the proposed mechanism to diverse membrane remodeling processes would be too speculative at this point.

2) Fig 1 FRET experiments:

Amphipathic membrane insertions, which generate positive contributions in membrane curvature will increase the distance between donors and acceptors bound to the polar heads of lipids as compared to dyes placed at the end of lipid acyl chains

FRET pair lipid heads: TopFLuor PE (donor) / Rhod PE/ Please describe this FRET pair (Fluorescein – Rhodamine? Introduce the spectral properties and why it was chosen as a good FRET pair.

FRET pair acyl chain: (TopFluor PC / TMR) Please justify the FRET couple as above. Explain in both cases how different photobleaching coming from the donor and/or the acceptor might affect your results.

-How did you measure the FRET efficiency. Do you mean apparent FRET efficiency, as authors utilized an intensity-based method?

When calculating the apparent FRET efficiencies in the presence/absence of Myomerger were these measurements carried out with low light levels to avoid photobleaching.

Do you think that the concentration (ratio of donor and acceptor fluorophores) played a role in your experiments? Did you try different concentrations of donor and acceptor to check for reproducibility?

We revised the description of the FRET experiments in the Methods to address the questions raised by the Reviewer. We modified figure Fig. 1A for FRET assay to present E_{FRET} in a more conventional way as equal $1 - I(\text{DA})/I(\text{D})$ rather than as $I(\text{DA})/I(\text{D})$. We have added emission and absorption spectra for all 4 fluorescent

probes (Fig. S8). Emission spectra were measured in liposomes with dyes incorporated separately at 0.5 mol % concentration. We measured absorption spectra for dyes dissolved in methanol. Using absorption and emission spectra, we calculated R_0 of 5.7 nm for TopFluor PE/Rhod PE FRET pair and 5.8 nm for TopFluor PC / TMR PC. The choice of the fluorescently labeled lipids, especially probes in the hydrocarbon tail position, is limited, and the dye's commercial availability was an important consideration. Another essential factor is photostability of the dyes, since photobleaching would affect measured FRET efficiency. In the conditions of our experiments, we detected no photobleaching of either of the used dyes.

Our setup allows us to measure FRET efficiency equivalent to the ratio of acceptor lifetime in the presence of acceptor to that in the absence of acceptor. The main difficulties in using fluorescence intensity for FRET efficiency measurement are (i) the spectral overlap of emission spectra of donor and acceptor and (ii) direct excitation of the donor by incident light. We accounted for the first effect using spectral de-mixing and for the second using control experiments with an infinite dilution of dyes by the addition of excess detergent. Details of the fitting procedure are described in the Methods.

We performed our experiments at a single ratio of donor/acceptor probes. It is possible that other ratios or probe concentrations can increase the sensitivity of the method, but our approach measures small relative changes and our conclusions should not depend on the specific donor and acceptor concentrations. Extensive optimization of the method is in our plans, but we believe it is outside of this paper's scope.

3) Experimental verification of the predictions of the model:

Cell-Cell fusion experiments performed with HA-cells and bound RBC.

-Addition of LPC before hemifusion arrests fusion reaction whilst cells in which hemifusion occurred (different concentrations, lower) induce fusion pore formation.

High concentrations of Myomerger right after hemifusion will also suppress fusion.

To further validate the hypothesis of low concentrations of myomerger playing an important role to drive hemifusion to fusion in the cell-cell experiments one could think of an experiment in which myomerger is labelled (not sure if a fluorescent protein might kill function here) and once could assess the fusion process as a function of time (from hemifusion to fusion pore formation and stabilisation or flickering), see procedure in [Jones et al., 2017 Cell Rep]. If labelling of the myomerger is not possible (even in the intracellular domain), perhaps one could fix the cells after the cell-cell kinetic experiments and perform immunostaining. This way one might find a relationship between myomerger expression and fusion (in all steps).

We agree that correlating the levels of Myomerger (and other proteins involved) expression and time course of myoblast fusion progression “from hemifusion to fusion pore formation and stabilisation or flickering” is important. However, we

consider this project and even measuring local concentrations of Myomerger at the sufficiently-well-resolved time in the vicinity of reliably distinguishable fusion intermediates to be an ambitious major project. In the revised manuscript we have addressed more limited questions. We have found that fluorescently-tagged synthetic Myomerger ectodomain rescues fusion of Myomerger-deficient C2C12 cells similarly to unlabeled synthetic Myomerger peptide (Fig. 10). Using this fluorescent peptide, we estimated membrane concentrations of Myomerger ectodomain that promote hemifusion-to-fusion transition of myomerger-deficient cells as ~ 500 molecules/ μm^2 . We also used fluorescent peptide to verify that Myomerger readily enters the tight and extended cell-cell contact regions and thus can be expected to be present in the vicinity of hemifusion connections (Fig. S9). (see our Reply to 2nd comment of the Reviewer 2).

Reviewer #2 (Remarks to the Author):

In this manuscript the authors provide a theoretical mechanism for the observation that a specific protein, myomerger, promotes fusion in myoblast cells.

The proposed mechanism assumes that myomerger proteins, which adsorb at the proximal leaflets of the fusing membranes, induce stress in the hemifusion diaphragm and thereby promoting opening of the full fusion pore.

The plausibility of the theory is, on the one hand, supported by calculations showing how reasonable choices for the parameters involved indeed lead to a speed-up in fusion rate, and, on the other hand, by supporting experiments showing that (i) indeed myomerger promotes positive curvature to a similar extent as lysolipids such as LPC, and (ii) addition of myomerger, as well as LPC, can rescue cells that are trapped in hemifused states.

Despite the overall elegance of the proposed theory and the apparent support from the experimental measurements, I am not convinced that we now understand the mechanism through which myomerger acts, for the following reasons:

1) First of all, the proposed mechanism seems to be very generic. According to the theoretical model, any membrane surface active compound (albeit peptide or protein, or anything that embeds at the membrane surface) would promote opening of the fusion pore. In the current study, this has been shown for LPC, but I would like to see more examples, and, in particular, are there negative controls?

We agree with the Reviewer that the mechanism we propose is rather generic. The molecular details of myomerger/membrane interactions and mechanisms that control the concentrations of myomerger at the time and place of fusion remain to be understood. Therefore, in our study we focus on the physical essence of the myomerger function. Our model suggests that not only myomerger and LPC but any membrane surface active compound that generates positive spontaneous curvature of the lipid monolayer would promote opening of a fusion pore. In original version of our paper, we noted that magainin 2, known to impose positive

curvature strain ⁵ promotes fusion pore formation for myomerger-deficient myoblasts ⁶. In revision, to address this question of the Reviewer, we found another positive curvature-generating peptide melittin ⁷ to also promote fusion pore opening (new Fig. 9). In contrast, a negative curvature generating peptide penetratin ⁷ and lipid oleic acid ⁸ did not promote pore formation in myomerger-deficient cells (Fig. 9 and Fig. S5A,B). These new experiments further substantiate our hypothesis that myomerger-driven fusion pore opening depends on the myomerger ability to generate positive curvature.

2) It remains unclear how the myomerger proteins could adsorb in the vicinity of the HD to exert their curvature-induced stress, as the HD is surrounded by the proteins of the fusion machinery which keep the fusing membranes together.

Since ~6kDa myomerger is much smaller than ~150,000 kDa antibodies routinely used to stain proteins in the cell-cell junctions, we expected it to readily enter the junctions. In our new experiments we directly verified this using fluorescent TAMRA-sMyomerger²⁶⁻⁸⁴ peptide and confocal microscopy. We have focused on the very tight contacts between HA-cell and bound RBC mediated by HA interactions with its sialic acid receptors. These cells are known to establish the extended contact with an area on the order of tens of square microns that are characterized by a relatively constant intermembrane distance of ~13 nm ⁹. Fluorescence labeling was observed throughout the contact zone within 5 min after application of the peptide to pre-bound cells (Fig. S9) indicating that myomerger rapidly diffuses into the tight contact zone.

3) The theoretical calculations depend on many parameters. As the authors acknowledge, most of these parameters are not well known and hard to measure in the first place, even for single component lipid systems. In vivo, we are dealing with a much more complicated situation with many different lipids and other proteins involved. For instance, the authors argue that "Since the spontaneous curvature of the distal monolayers remains unchanged, this can happen only at the expense of deviation of the distal monolayers from their optimal configurations within the junction region." However, a change in lipid composition could be another (and energetically less costly) way to induce changes without the need for elastic stresses building up in the HD. Such changes could occur on very fast time scales (requiring only local lipid redistributions). In addition, the role of lipid flipflops (notably cholesterol) should be discussed in potentially affecting the release of stress.

We agree that the lipid redistribution between the membrane bulk and the elastically stressed regions of the membrane monolayers, generally, reduces the elastic energy and affects the stress-related phenomena. For the situation considered in this work, the regions of lipid monolayers forming the HD rim are characterized by a substantially negative splay of lipid molecules. Therefore, depletion of lipid molecules with positive molecular curvature from or enrichment of lipid molecules with negative molecular curvature in the HD rim region would decrease the local stress and, consequently, reduce, to some extent, the acceleration factor of the fusion pore formation. To evaluate this effect, one has to consider that the lipid enrichment and/or depletion in certain regions are counteracted by the entropy, which favors an even distribution of all lipid species

across the whole membrane plane. Therefore, the lipid repartitioning is expected to be substantial only if the resulting relaxation of the elastic energy related to one lipid molecule exceeds the thermal energy $k_B T$ (the product of the Boltzmann constant and the absolute temperature), which sets the scale of the entropic energy penalty per molecule. Given the smallness of the area occupied by one lipid molecule in the membrane plane, the elastic energy relaxation is, typically, smaller than $k_B T$, so that the effects of the lipid redistribution are small.

This said, we estimated the effect of the lipid redistribution on the acceleration factor predicted by our model. As a relevant example we considered the depletion from the rim region of LPC molecules characterized by a large molecular curvature, $\xi_{LPC} = 0.26 \text{ nm}^{-1}$. Following a similar derivation procedure as in ¹⁰ we found that the local mole fraction of LPC, ϕ_{LPC}^p , within the rim region characterized by average local lipid splay, \tilde{J} , is related to the mole fraction of this lipid in the membrane bulk, ϕ_{LPC}^r , by,

$$\phi_{LPC}^p = \frac{\phi_{LPC}^r}{(1 - \phi_{LPC}^r) \exp\left(-\kappa_m \tilde{J} a \frac{\xi_{LPC} - \xi_0}{k_B T}\right) + \phi_{LPC}^r},$$

where a is the area per lipid molecule in the membrane plane, which we assumed identical for all lipids, ξ_0 is the molecular curvature of the background lipid, and κ_m is the bending modulus of a lipid monolayer. For estimation, we use the typical parameter values, $\kappa_m = 10 k_B T$, ¹¹, $\xi_0 = -0.1 \text{ nm}^{-1}$ ^{12,13}, $a = 0.7 \text{ nm}^2$ ¹¹⁻¹³. For the sake of a conservative estimation, we used the upper limit of the considered range of LPC bulk mole ratios, $\phi_{LPC}^r = 0.1$. According to our calculations, the splay in the proximal monolayer has the most negative value near the HD rim and decays to negligible values within a distance of a few nanometers. Assuming the characteristic decay length of deformations to equal $l = 1.5 \text{ nm}$, we calculated the mean splay to be $\tilde{J} \sim -0.1 \text{ nm}^{-1}$. Using this value for \tilde{J} , and assuming the typical geometrical characteristics of the fusion site to be $R_B = 15 \text{ nm}$, $x^* = 6.6 \text{ nm}$, the pore line tension and the saddle splay modulus to constitute $\lambda = 10 \text{ pN}$ and $\bar{\kappa}_m = -5 k_B T$, respectively, we found the local mole fraction of LPC around the rim to be $\phi_{LPC}^p \sim 0.08$. The acceleration factor was computed to be reduced from $\beta = 8.3$ before LPC redistribution to $\beta = 6.9$ after the redistribution. This means that both the local LPC concentration in the rim region and the acceleration factor decreased by about 20%. Such correction does not change the qualitative conclusions of our work. We have included this estimate and the related comment into the text of the manuscript.

Similarly, one could expect a fast flip-flop-driven redistribution between the membrane monolayers of such molecules as cholesterol to relax the elastic stress and, hence, affect the acceleration factor. However, it has to be appreciated that the monolayer area involved in formation of the fusion site, which includes HD and its rim, is small compared to the area of the surrounding membrane. This means that the monolayers of the plasma membrane play a role of lipids reservoirs for the corresponding monolayers of the fusion site and set the compositions of these

monolayers. The lipid flip-flop in the fusion site, if happening, is expected to be buffered by the molecule exchange with the reservoirs and, therefore, to have no effect. We included the related comment into the text of the revised manuscript.

4) It also remains puzzling what concentrations are needed to promote fusion. It is stated that LPC can promote fusion pore opening (from cells that are trapped in a hemifused state) most efficiently at 50 microM, and that 300 microM is needed to block hemifusion. But what happens to cells that are treated with 50 microM before hemifusion takes place ? Is full fusion still observed or not ? The same question arises wrt myomerger.

In the experiments on HA-mediated fusion (presented in Fig. 4, 5 and Fig. S1) we applied LPC and myomerger to HA-cell/RBC complexes already after the end of low pH pulse. We selected this experimental design because it allowed us to exclude contributions of any interactions of either myomerger or LPC at low pH with either membranes or transient low pH conformations of HA. In biological context of myoblast fusion such low pH-dependent interactions of myomerger (if any) would likely be irrelevant.

To address this comment of the Reviewer, we applied myomerger ectodomain to HA-cell/RBC complexes before application of low pH pulse. Low concentration of myomerger promoted complete fusion and high concentration inhibited both lipid and content mixing (Fig. S2). This result is consistent with our conclusions that (i) at high concentrations myomerger inhibits hemifusion and complete fusion and (ii) there is a range of concentrations in which myomerger does not block hemifusion and promotes fusion pore opening. We have not carried out similar experiments with LPC because, in contrast to myomerger that tightly binds to membranes, LPC readily redistributes between membranes and LPC-free medium. So, we would have to include LPC into the neutral pH medium before low pH pulse, then into the low pH medium, and then into the neutral pH medium applied at the end of low pH pulse. This increases cytotoxic effects of LPC and likelihood of content probe leakage from RBCs.

We have included the new figure (Fig. S2) and its discussion into the revised MS.

5) The explanation of the results from the cell experiments are difficult to interpret, which is not surprising given the complexity of cell fusion. The manuscript contains some speculative text on why certain experimental findings are not in line with the predictions from the theory (e.g., arguments about why synchronized cells show efficient fusion in absence of LPC or myomerger, arguments about the relevance of a 4% speedup, arguments about why high concentrations block fusion, etc.). The manuscript would be strengthened with more convincing explanations or additional experiments on these topics.

To better explain our interpretation for the lack of the fusion promotion at higher concentrations of LPC (Fig. 6D), we added the following sentences: “The lack of the promotion for higher concentrations of LPC and sMyomerger²⁶⁻⁸⁴ can be explained if hemifusion connections that did not advance to fusion pores

dissociate. If hemifusion connections appear and disappear continuously, the high concentrations of LPC and sMyomerger²⁶⁻⁸⁴ are expected to block appearance of new hemifusion connections but facilitate fusion for the cells that are already hemifused at the time of LPC/sMyomerger²⁶⁻⁸⁴ application. In contrast, application of LPC/sMyomerger²⁶⁻⁸⁴ concentrations that are low enough to allow hemifusion will, in addition, promote to fusion all new hemifusion connections developing during 30-60 min in the presence of the reagents. “

To better explain our interpretation for the finding that even without sMyomerger²⁶⁻⁸⁴ application, the levels of fusion of myomerger-deficient cells in the LPC-synchronization experiments were higher than those observed in the unsynchronized fusion of these cells we included additional experiments and added/edited the following sentences: “Note that even without sMyomerger²⁶⁻⁸⁴ application, the levels of fusion of Myomerger-deficient cells in the LPC-synchronization experiments were higher than those observed in the unsynchronized fusion of these cells. This finding suggested that LPC application converts the existing hemifusion connections into complete fusion and prevents formation of new hemifusion intermediates during 16-hour incubation of the cells in the presence of LPC. This interpretation has been supported by finding that shortening the time interval between co-plating of the cells and LPC application expected to lower the numbers of hemifused Myomerger-deficient cells lowered the extents of fusion observed after LPC removal (Fig. S3).”

6) Even if we take for granted that addition of myomarker or LPC in hemifused cells indeed have a positive effect on fusion pore opening, to me it seems that ANY type of shock you add to the cell at this stage would have such an effect. After all, the hemifused state is likely a kinetically trapped state requiring some form of energetic input to be released. Are there any negative controls ?

In our new experiments we found that, in contrast to positive spontaneous curvature generating peptides/lipids myomerger, LPC, melittin (and magainin 2 in our earlier work ⁶), negative spontaneous curvature generating peptides/lipids (penetratin and oleic acid) did not promote opening of a fusion pore in myomerger-deficient myoblasts. Moreover, application of oleic acid inhibits hemifusion-to-fusion transition in w.t. myoblasts (and influenza HA- mediated fusion in our earlier work ¹⁴). These negative controls argue against the hypothesis that any additions promote fusion pore opening.

7) To bridge the gap between the theory on one side, and the full cell experiments on the other side, what I miss in this study are experiments performed on liposomal fusion assays. Here, many of the predictions from the theory could be much more straightforwardly tested.

Following the reviewer's suggestion, we performed experiments on protein-free liposomes but, unfortunately, have not been able to reproduce our findings on cells in liposomes. In our experiments, we used cell-size giant unilamellar vesicles, GUV). It was previously shown that PS-containing GUVs in the presence of ~ 2mM Ca²⁺ could be arrested at the extended hemifusion stage (Nikolaus et al., Biophys. J. 2010: 98, 1192-1199). We successfully reproduced the assay and observed

hemifusion using GUVs formed by swelling on PVA gels (Fig. R1). To more robustly detect full fusion events, we slightly modified the approach by incorporating

Figure R1. **Ca²⁺ induced hemifusion of PS containing GUVs.** GUVs containing red lipid (RhDOPE) and content (Alexa Fluor™ 568 Hydrazide) dyes were induced to hemifuse with GUVs containing green lipid (TopFluorDOPE) and content (Alexa Fluor™ 488 Hydrazide) dyes in presence of 2mM CaCl₂. GUV were formed by PVA gel assisted swelling. Liposomes were formed from 39.5:30:30:0.5 DOPC:DOPE:DOPS:LipidDye lipid mixture, where Lipid Dye was either RhDOPE (red) or TopFluorDOPE (green). Liposomes were prepared in buffer containing 280 mM Sucrose, 10 mM HEPES, pH =7.0 and 5uM of either Alexa Fluor™ 568 Hydrazide (for liposomes containing RhDOPE) or Alexa Fluor™ 488 Hydrazide (for liposomes containing TopFluorDOPE). Differently labeled liposomes were mixed together and transferred to BSA coated (to prevent spreading of liposomes on glass) glass bottom 35mm petri dishes containing 300 mM Glucose, 10 mM HEPES, pH=7.0. Hemifusion was induced by addition of CaCl₂ to achieve 2mM CaCl₂ concentration. Hemifused liposomes could be observed already within few minutes after Ca²⁺ addition.

content probes (Alexa488 and Alexa 568) into the GUVs. We observed almost no complete fusion events and application of 50 to 200 μM of lauroyl LPC had not increased the probability of complete fusion. We hypothesize that this distinction from our observations on cells could be due to a high mole fraction of phosphatidylethanolamine (PE), a lipid inducing a monolayer's negative spontaneous curvature that we had to include to observe hemifusion in this system. As our model predicts, more negative spontaneous curvature of proximal monolayer and higher line tension of initial pore (due to PE in distal monolayers) significantly increase the energy barrier for fusion pore opening. Furthermore, in yet unexplained distinction, GUVs tend to form huge GUV-sized hemifusion structures¹⁵ that have never been observed for biomembranes. The larger area of the fusion site is also expected to increase the barrier's energy. Thus, we may be unable to reach LPC concentrations necessary to promote a transition from hemifusion to complete fusion in this system.

It is also possible that PS-Ca²⁺-PS bridges that hold bilayer together generate so tight contacts that hemifusion structure developing within such contacts are inaccessible for LPC/myomerge.

We agree with the reviewer that the protein-free systems could provide valuable new insights into the hemifusion-fusion transition mechanism, but we believe the development of the appropriate model system is a major project and is beyond

this work's scope.

8) At the end of the manuscript the potential role of fusion peptides is mentioned. According to the authors, such fusion peptides (which are also known to be embedded at the membrane surface) could also promote fusion pore opening via the same mechanism. In this light, simulation studies show indeed a mechanism by which fusion peptides can induce fusion pores, namely via promoting formation of stalk-pore complexes. It would be nice to cite these studies and discuss whether the creation of stalk-pore complexes could play a role here as well. See for instance work by Risselada and colleagues, and Fuhrmans & Marrink.

Thanks. We have added a sentence mentioning these alternative models. “Other mechanisms of promotion of fusion pore formation by membrane-interacting regions of viral fusogens were modelled by molecular simulations of pore-stalk complexes ^{16,17} “

Reviewer #3 (Remarks to the Author):

In this manuscript, Golani et al. investigated how the fusogenic peptide myomerger facilitates the transition from hemifusion stalk to fusion pore formation during membrane fusion. They hypothesized that myomerger generates a positive spontaneous curvature of the proximal monolayer. The membrane elastic stresses then propagate into the hemifusion diaphragm, which accelerates fusion pore formation. Interestingly, the effect of myomerger ectodomain is strikingly similar to that of LPC, which is known to generate a positive spontaneous curvature of lipid monolayers, further validating the hypothesis.

This study aims to fill a major gap in our understanding of membrane fusion, namely how a hemifusion stalk resolves into a fusion pore. The theoretical modeling is effective in showing that positive curvature of the proximal monolayers generates additional stress in the hemifusion diaphragm and accelerates fusion pore formation. Experiments in cells largely validated the model. Overall, this is a very interesting study that provides novel insights into a critical step during membrane fusion.

Major comments:

1. The authors showed that both LPC and sMyomerger (at low concentration) facilitate the transition from hemifusion stalk to fusion pore formation, likely by inducing positive curvature of the proximal monolayer. To support this conclusion, the authors should test whether a lipid that generates negative curvature would inhibit this transition.

In our earlier work we have reported that application of a negative-curvature-generating lipid OA to HA-cell/RBC pairs promotes hemifusion and inhibits the hemifusion-to-fusion transition ¹⁴. In our new experiments we found that OA application to LPC-synchronized wild type C2C12 cells 5 min after LPC removal inhibits myoblast fusion. We included the results of these experiments into the revised manuscript (Fig. S5) and they, as suggested by the Reviewer, are consistent with our conclusions. We also added a comment that, in contrast to

LPC, OA added to the proximal monolayers of the membranes (and other lipids of negative curvature) generally quickly redistributes between membrane monolayers. Possible contributions of OA in the inner leaflets complicate the interpretation of these findings.

2. If high concentration of sMyomerger inhibits the transition from hemifusion stalk to fusion pore formation, does overexpressing a high level of wild-type Myomerger in myomerger-deficient cells inhibit myoblast fusion?

As noted by the Reviewer, we found high concentrations of sMyomerger²⁶⁻⁸⁴ to inhibit w.t. myoblast fusion (Fig. 7D,E). We now tested whether overexpressing full-length wild-type Myomerger also inhibits fusion of w.t. C2C12 cells. Intriguingly, we found that a robust overexpression of Myomerger, confirmed by an increase of its content in the cell lysate, was not accompanied by a detectable increase of Myomerger content on the plasma membrane, evaluated by the surface biotinylation analysis (Fig. S4). While this finding precludes us from testing whether raising membrane concentration of full-length Myomerger beyond its normal level, similarly to high concentration of sMyomerger²⁶⁻⁸⁴, inhibits fusion, it also suggests the existence of a yet-unexplored mechanism that controls trafficking of Myomerger to/from plasma membrane. We plan to explore this mechanism but consider this extension to be beyond the scope of this paper. In the revised MS we present the new figure. In Discussion we mention that a regulatory mechanism that, as suggested by these data, controls the plasma membrane localization of Myomerger may preclude any potential ability for Myomerger to inhibit hemifusion and fusion in myoblasts by never allowing this protein to reach hemifusion-inhibiting membrane concentration.

3. Based on the previous study (Leikina et al., 2018), myomerger-deficient cells should be stalled at the hemifusion stage. After the incubation scheme in Figure 6, what was the percentage of myomerger-deficient cells that have formed hemifusion stalks? How long would the potential fusion partners stay attached by the hemifusion stalks?

See below.

4. After releasing the LPC block in Figure 7, did the percentage of cells linked by hemifusion stalks increase, by how much? Can all of these potential fusion partners linked by hemifusion stalks proceed with fusion in the presence of low concentrations of LPC and sMyomerger? How fast is each fusion event? Why would high concentrations of sMyomerger not inhibit fusion after releasing the LPC block?

Reply to 3 & 4. Hemifused cells are conventionally identified as cells that had exchanged lipid probes but had not exchange content probes. Earlier studies suggested that hemifusion events in lipid bilayer fusion, HA-mediated fusion and SNARE-mediated fusion¹⁸⁻²⁰ are reversible. The membranes that experienced transient hemifusion exchange lipid probes but then lose hemifusion connections. The operational definition above (lipid mixing without content mixing) does not distinguish cells that are connected by hemifusion connections at the time of analysis from the cells that had been connected by such connections at some time

before the analysis, but these connections had already dissociated. Earlier work has validated an alternative approach of detection of currently present hemifusion connections by application of chlorpromazine (CPZ)^{6,14,19,21}. In this approach cell-cell contacts with hemifused connections are revealed by their conversion into complete fusion (content mixing). Using this approach in our new experiments we have for the first-time established reversibility of hemifusion in myoblast fusion (and in any developmental cell fusion process). To characterize the lifetime of hemifusion connections in myomerger-deficient C2C12 cells we accumulated differentiating cells with LPC upstream of hemifusion for 16h and then removed LPC to allow hemifusion/fusion. Release from the LPC block temporary increased the numbers of hemifused cells. Immediately and at different times after LPC removal we applied 50 μ M CPZ for 1-min to reveal the existing hemifusion connections by their conversion into full fusion. The data indicated that the percentage of hemifused cells decreased two-fold from \sim 8% to a background level of \sim 4% within 30 min. A limited lifetime of hemifusion connections suggests that differentiated myomerger-deficient myoblasts maintain a background level of hemifusion by constantly forming transient hemifusion connections. These experiments give us rough estimates of the characteristic time of dissociation of hemifusion connections and the percentage of myomerger-deficient cells that have had hemifusion connections at the time of CPZ application. This data also suggests that the answers to the Reviewer questions: “After the incubation scheme in Figure 6, what was the percentage of myomerger-deficient cells that have formed hemifusion stalks?” and “After releasing the LPC block in Figure 7, did the percentage of cells linked by hemifusion stalks increase, by how much?” are at least 4% and at least 4%, respectively. More accurate estimates of the hemifusion extents at different time points depend on the unknown efficiency with which CPZ converts hemifusion into detectable full fusion. We also do not know the efficiency with which application of low concentrations of LPC and sMyomerger converts hemifusion connections into full fusion and, thus, cannot experimentally answer the question: “Can all of these potential fusion partners linked by hemifusion stalks proceed with fusion in the presence of low concentrations of LPC and sMyomerger?”.

“How fast is each fusion event?” This interesting question should probably be addressed by electrophysiology allowing very fast detection of smallest fusion pores. We consider it to be beyond the scope of our work.

“Why would high concentrations of sMyomerger not inhibit fusion after releasing the LPC block?” We suggest that application of sMyomerger to the synchronized cells does not inhibit hemifusion/fusion because, based on the experiments presented in Fig. 7, by the time LPC is removed by 3 washes with LPC-free DM hemifusion (this takes 1-2 min) hemifusion extents already reach maximum.

Minor comments:

1. In the HA-RBC fusion experiments, lipid mixing (indicating hemifusion) seems to be much higher compare to a previous study (Leikina et al., 2018). How to explain the difference?

In ⁶ we have used 1 µg/ml trypsin, 2 minutes at room temperature and in this study, we treated HA-cells with 1 µg/ml trypsin for 5 minutes at room temperature. The reason for extending the trypsin application and, thus, somewhat increasing the number of fusion competent HA1-HA2 is that while in Leikina et al., 2018 we have been focused only on promotion of fusion, here we needed to also look for inhibition and, thus, preferred to start with higher levels of lipid mixing.

2. Does HA activation change the percentage of HA-cell/RBC pairs, which in turn could partially explain the increased number of full fusion events?

Since unbound RBC are washed out before low pH, LPC, sMyomerger26-84 applications, HA activation is not expected to change the percentage of HA-cell/RBC pairs. To verify this, we compared for different conditions the numbers of cell-associated RBCs per field N_{ca} , defined as the sum of the number of unfused RBCs N_{uf} (RBCs bound but not fused to HA-cells detected as labeled RBC bound to unlabeled HA cell) and the number of fused/hemifused RBCs per field $N_{f/hf}$ (scored as the number of HA-cells that acquired content- and/or membrane probes). For instance, for Fig. 4B, N_{ca} for 50µM (30.2+/- 1.4) vs. 0µM LPC (29.5+/-1.9) (n=2). Finding that low pH and fusion pore promoting concentration of LPC did not change N_{ca} confirmed that changes in fusion/hemifusion efficiencies could not be explained by changes in the efficiency of HA-cell/RBC docking.

3. In Figure 4C, there are few bright green cells without red signal in the 50 µM LPC-treated cells. What are these cells?

Figure R2. As a result of complete fusion, between HA-cells and RBC labeled with red membrane probe PKH26 and green content probe carboxyfluorescein RBC acquired both green and red probes. Because of heterogeneous labeling of RBCs with different probes, some HA cells, including the one shown by arrows are strongly labeled with CF (bright green) and a weaker PKH26 labeling (weak red). On top, the image from the original version of the MS. Two images on the bottom show green fluorescence (left) and red fluorescence (right).

These are HA-cells that as a result of complete fusion with RBC acquired green content probe (carboxyfluorescein (CF). Labeling of RBCs with red membrane probe PKH26 and CF is rather heterogenous. HA-cell fusion with RBC that happen to be strongly labeled with CF and a weaker PKH26 labeling generates bright green HA-cell with only weak PKH26 labeling (Fig. R2). Note that regardless of the intensity of PKH26 labeling we scored green HA-cells as complete fusion events. In the revised MS we replaced this image with the

image where green and red labeling are better balanced.

4. In Figure 4D, the control showed an average of 70% lipid mixing and less than 10% content mixing. With the same setting, Figure 5A showed similar lipid mixing, but significant more content mixing (>40% vs. 10% in Figure 4D). Given that these were similar experiments, it is difficult to appreciate the inhibitory effect driven by sMyomerger if the control content mixing can be as low as 10%.

Thank you for drawing our attention to this. Fusion extents and, especially, fusion/hemifusion ratio in the suboptimal conditions are very sensitive to relatively minor changes in HA expression (see for instance ¹⁴). As a result, for different batches of cells and on different days, treating HA-cell/RBC pairs with the same pulses of low pH (say 1 min application of pH 5) resulted in different efficiencies of hemifusion/fusion. As noted in the Methods, “fusion extents and kinetics varied from day to day, apparently as a result of variation in the level of HA expression, we routinely started the experiments by choosing the precise conditions of the low pH treatment.” In all cases, all the data presented in one panel of the figures for different conditions (with or without LPC/myomerger applications) have been gathered in the parallel experiments carried out at the same time. The experiments presented in Fig. 4D and Fig. 5A were carried out at different times on different batches of cells, and the same low pH treatment (1 min application of pH 5 medium) resulted in different fusion efficiencies. In the revised manuscript we added an additional comment on this point.

5. In Figure 5A, 2.5 μ M sMyomerger seemed to significantly promote lipid mixing compared to the control, but did not enhance content mixing. Could the authors provide some explanations for this?

We think that under these conditions sMyomerger promotes formation of small and, possibly, short-living fusion pores that facilitate exchange of membrane probe but do not allow sufficient redistribution of a content probe to detect it in HA-cells. A retardation of the aqueous probe redistribution relative to membrane probe redistribution for small fusion pores in HA-cell/RBC fusion has been documented in ²².

6. The statistical differences in Figure 6B and 6D are relatively small, even when the fusion index was counted as any cells with more than 1 nucleus. It is necessary to make sure by live imaging that the binucleate cells did not result from incomplete cytokinesis.

At the time we score fusion, differentiated C2C12 already do not divide. In the new analysis carried out in revision, we counted as fused cells only double labeled cells plus cells that are not double labeled but have more than 2 nuclei. Cells with two nuclei labeled with only one of the probes were not included. The findings from this new analysis parallel the findings in which we counted as fused all cells with ≥ 2 nuclei/cell (compare Fig. S7 vs Fig. 7 and Fig. S3D vs Fig. S3B).

7. In Figure 7B, the myomerger-deficient cells were allowed to differentiate for 72 hrs before mixing, whereas the WT cells in Figure 7C only differentiated for 48 hrs and allowed to fuse for 4 hrs. Why did the authors use much longer differentiation time for the myomerger-deficient cells? Due to the different experimental conditions, it is difficult to compare the levels of fusion between WT cells and myomerger-deficient cells synchronized to fuse.

We used a shorter differentiation period in the experiment shown in Fig. 7D (now Fig. 8D) than in all other experiments (~2 days of incubation the differentiation medium vs. ~3 days in all other experiments) because we wanted to limit the duration of application of the Myomerger ectodomain by applying it at the time of the most robust fusion. At this time the changes in the fusion efficiency have stronger effects on measured fusion extents.

Indeed, the extents of fusion in Fig. 7B (synchronized, now Fig. 8B) and 7D (unsynchronized, now Fig. 8D) should not be compared because, as noted by the Reviewer, they correspond to different time points ~2 days (48+4 hours) in D vs. 3 days in B. In the revised MS, we have now an experiment (Fig. S5C), in which we have the extent of synchronized fusion of wild type C2C12 cells after 3 days of differentiation of ~15%. Similar range of fusion extents in synchronized fusion of w.t. C2C12 after 3 days of differentiation have been reported in ^{23,24}. We have added a comment on this comparison to the text of the revised MS.

8. In the method section, the timelines for myoblast fusion are somewhat confusing. Why were the cells plated in differentiation medium for ~85 hours? Also, the fixation time should be indicated differently for different experiments (not ~ 85 hr post differentiation for all experiments), e.g. the WT C2C12 cells were only allowed to differentiate for 48 hrs compared to the 72 hrs differentiation time for the myomerger-deficient cells.

We very much appreciate this critique and apologize for the confusing descriptions. We revised the explanations of the timelines for myoblast fusion experiments in the Methods and figure legends to clarify/correct the descriptions. We added a supplemental figure (Fig. S6) to summarize the timelines in all experimental designs used in our myoblast fusion experiments.

References.

- 1 Kozlovsky, Y. & Kozlov, M. M. Membrane fission: model for intermediate structures. *Biophys J* **85**, 85-96, doi:10.1016/S0006-3495(03)74457-9 (2003).
- 2 Mattila, J. P. *et al.* A hemi-fission intermediate links two mechanistically distinct stages of membrane fission. *Nature* **524**, 109-113, doi:10.1038/nature14509 (2015).
- 3 Zhao, W. D. *et al.* Hemi-fused structure mediates and controls fusion and fission in live cells. *Nature* **534**, 548-552, doi:10.1038/nature18598 (2016).

- 4 Zhang, G. & Muller, M. Rupturing the hemi-fission intermediate in membrane fission under tension: Reaction coordinates, kinetic pathways, and free-energy barriers. *J Chem Phys* **147**, 064906, doi:10.1063/1.4997575 (2017).
- 5 Matsuzaki, K. *et al.* Relationship of membrane curvature to the formation of pores by magainin 2. *Biochemistry* **37**, 11856-11863, doi:10.1021/bi980539y (1998).
- 6 Leikina, E. *et al.* Myomaker and Myomerger Work Independently to Control Distinct Steps of Membrane Remodeling during Myoblast Fusion. *Dev Cell* **46**, 767-780 e767, doi:10.1016/j.devcel.2018.08.006 (2018).
- 7 Koller, D. & Lohner, K. The role of spontaneous lipid curvature in the interaction of interfacially active peptides with membranes. *Biochim Biophys Acta* **1838**, 2250-2259, doi:10.1016/j.bbamem.2014.05.013 (2014).
- 8 Chernomordik, L. V. & Kozlov, M. M. Protein-lipid interplay in fusion and fission of biological membranes. *Annu Rev Biochem* **72**, 175-207, doi:10.1146/annurev.biochem.72.121801.161504 (2003).
- 9 Leikina, E. *et al.* Influenza hemagglutinins outside of the contact zone are necessary for fusion pore expansion. *The Journal of biological chemistry* **279**, 26526-26532, doi:10.1074/jbc.M401883200 (2004).
- 10 Choudhary, V. *et al.* Architecture of Lipid Droplets in Endoplasmic Reticulum Is Determined by Phospholipid Intrinsic Curvature. *Curr Biol* **28**, 915-926 e919, doi:10.1016/j.cub.2018.02.020 (2018).
- 11 Fuller, N. & Rand, R. P. The influence of lysolipids on the spontaneous curvature and bending elasticity of phospholipid membranes. *Biophysical Journal* **81**, 243-254 (2001).
- 12 Szule, J. A., Fuller, N. L. & Rand, R. P. The effects of acyl chain length and saturation of diacylglycerols and phosphatidylcholines on membrane monolayer curvature. *Biophys J* **83**, 977-984, doi:10.1016/s0006-3495(02)75223-5 (2002).
- 13 Chen, Z. & Rand, R. P. The influence of cholesterol on phospholipid membrane curvature and bending elasticity. *Biophys J* **73**, 267-276, doi:10.1016/S0006-3495(97)78067-6 (1997).
- 14 Chernomordik, L. V., Frolov, V. A., Leikina, E., Bronk, P. & Zimmerberg, J. The pathway of membrane fusion catalyzed by influenza hemagglutinin: restriction of lipids, hemifusion, and lipidic fusion pore formation. *J Cell Biol* **140**, 1369-1382, doi:10.1083/jcb.140.6.1369 (1998).
- 15 Nikolaus, J., Stockl, M., Langosch, D., Volkmer, R. & Herrmann, A. Direct visualization of large and protein-free hemifusion diaphragms. *Biophys J* **98**, 1192-1199, doi:10.1016/j.bpj.2009.11.042 (2010).
- 16 Fuhrmans, M. & Marrink, S. J. Molecular view of the role of fusion peptides in promoting positive membrane curvature. *J Am Chem Soc* **134**, 1543-1552, doi:10.1021/ja207290b (2012).
- 17 Risselada, H. J. *et al.* Line-tension controlled mechanism for influenza fusion. *PLoS One* **7**, e38302, doi:10.1371/journal.pone.0038302 (2012).
- 18 Chanturiya, A., Chernomordik, L. V. & Zimmerberg, J. Flickering fusion pores comparable with initial exocytotic pores occur in protein-free phospholipid bilayers. *Proc Natl Acad Sci U S A* **94**, 14423-14428, doi:10.1073/pnas.94.26.14423 (1997).
- 19 Leikina, E. & Chernomordik, L. V. Reversible merger of membranes at the early stage of influenza hemagglutinin-mediated fusion. *Mol Biol Cell* **11**, 2359-2371, doi:10.1091/mbc.11.7.2359 (2000).

- 20 Giraudo, C. G. *et al.* SNAREs can promote complete fusion and hemifusion as alternative outcomes. *J Cell Biol* **170**, 249-260, doi:10.1083/jcb.200501093 (2005).
- 21 Melikyan, G. B., Brener, S. A., Ok, D. C. & Cohen, F. S. Inner but not outer membrane leaflets control the transition from glycosylphosphatidylinositol-anchored influenza hemagglutinin-induced hemifusion to full fusion. *J Cell Biol* **136**, 995-1005, doi:10.1083/jcb.136.5.995 (1997).
- 22 Zimmerberg, J., Blumenthal, R., Sarkar, D. P., Curran, M. & Morris, S. J. Restricted movement of lipid and aqueous dyes through pores formed by influenza hemagglutinin during cell fusion. *J Cell Biol* **127**, 1885-1894, doi:10.1083/jcb.127.6.1885 (1994).
- 23 Leikina, E. *et al.* Extracellular annexins and dynamin are important for sequential steps in myoblast fusion. *J Cell Biol* **200**, 109-123, doi:10.1083/jcb.201207012 (2013).
- 24 Gamage, D. G. *et al.* Insights into the localization and function of myomaker during myoblast fusion. *J Biol Chem* **292**, 17272-17289, doi:10.1074/jbc.M117.811372 (2017).

REVIEWERS' COMMENTS

Reviewer #2 (Remarks to the Author):

The authors have carefully addressed my concerns, in particular providing negative controls, giving much more credibility to the proposed mechanism. I am now more than happy to recommend the paper for publication !

Reviewer #3 (Remarks to the Author):

In the revised manuscript, the authors did a good job addressing most of my questions. Here are a few minor comments:

1. In addressing question 2, the authors used WT C2C12 cells to perform the overexpression experiment, and found that the surface concentration of myomerger is tightly regulated in these cells. What about overexpressing myomerger in myomerger-deficient cells?
2. The authors showed that applying OA to myomerger-deficient cells did not promote fusion. What about treating these cells first with OA and then with low concentration of LPC or sMyomerger? Does this increase fusion?
3. In Fig. 1A, the concentration of each data point needs to be indicated.

POINT-BY POINT RESPONSE

We are very grateful to the Reviewers for their thoughtful critiques/questions and for support that helped us to considerably strengthen our study.

REVIEWERS' COMMENTS

Reviewer #2 (Remarks to the Author):

The authors have carefully addressed my concerns, in particular providing negative controls, giving much more credibility to the proposed mechanism. I am now more than happy to recommend the paper for publication !

Thank you very much!

Reviewer #3 (Remarks to the Author):

In the revised manuscript, the authors did a good job addressing most of my questions. Here are a few minor comments:

1. In addressing question 2, the authors used WT C2C12 cells to perform the overexpression experiment, and found that the surface concentration of myomerger is tightly regulated in these cells. What about overexpressing myomerger in myomerger-deficient cells?

This comment focuses on the mechanism that controls surface levels of myomerger in myoblasts. We agree that it will be interesting to compare mechanisms that control cell surface expression of myomerger in myomerger-deficient myoblasts with those in WT myoblasts. We already know from published reports that over-expressing myomerger rescues fusion of myomerger-deficient cells. The question is whether in this case we can get the cell surface expression of myomerger to levels exceeding those observed for WT cells. We do not expect the mechanism that controls surface expression of myomerger to distinguish endogenous myomerger from exogenously expressed one. Even if this is not the case, in our opinion, to get interpretable results, this question has to be addressed in the context of a more general analysis of the mechanisms of regulation that we consider to be outside of the scope of the present work.

2. The authors showed that applying OA to myomerger-deficient cells did not promote fusion. What about treating these cells first with OA and then with low concentration of LPC or sMyomerger? Does this increase fusion?

The challenge with interpreting the data of such experiments will be related to a quick redistribution of OA between outer and inner leaflets of the membranes. As we mentioned in the text, “Note, however, that in the case of OA we cannot exclude possible contribution of OA in distal leaflets, since OA added to the proximal monolayers of the membranes, in contrast to LPC, quickly redistributes between membrane monolayers“. As a result, the expected effects of combined application of myomerger ectodomain and OA depends on the relative rates of OA redistribution and hemifusion dissociation, complicating definitive interpretation of such data. We respectfully suggest that these experiments have to be done in the context of a broader analysis of the rates of OA

redistribution and hemifusion dissociation.

3. In Fig. 1A, the concentration of each data point needs to be indicated.

Adding the concentration to each point on the Figure 1 A will unnecessarily crowd the figure. Since addition of the reagents leads to a monotonous decrease in E(FRET), reagent concentration decreases for the points from left of the figure to the right. We have made the changes in the figure legend to clarify this: “A) Increasing bulk concentrations of either dioleoyl phosphatidylcholine, DOPC (from 1 μM to 10 μM), or lysophosphatidylcholine, LPC (from 10 μM to 100 μM) or sMyomerger26-84 (from 50 nM to 0.5 μM) were added to 2.5 μM liposomes containing fluorescence resonance energy transfer (FRET) pair with both probes located in the lipid headgroup region or with both probes located in the lipid hydrocarbon tail region. Dependence of FRET efficiency characterized as change in quantum efficiency of donor emission due to the presence of acceptors (see Methods) for probes in headgroup region E(FRET) in Heads vs E(FRET) in Tails for probes in hydrocarbon tails from a representative experiment is shown. For all three reagents and for probes both in the headgroup region and hydrocarbon tails, E(FRET) monotonously decreased as the bulk concentration of the reagent increased. Individual points correspond to a single bulk concentration of the added molecules, line with the shading shows linear fit of the data with 0.95 confidence interval.”